



# Inconclusive Early warning signals for Dansgaard-Oeschger events across Greenland ice cores

Clara Hummel[1], Niklas Boers[2,3,4], and Martin Rypdal[1]

[1]Department of Mathematics and Statistics, UiT The Arctic University of Norway, Tromsø, Norway
[2]Earth System Modelling, School of Engineering & Design, Technical University of Munich, Munich, Germany
[3]Potsdam Institute for Climate Impact Research, Potsdam, Germany
[4]Department of Mathematics and Global Systems Institute, University of Exeter, Exeter, UK

**Correspondence:** Clara Hummel (c.hummel@uit.no)

**Abstract.** The Dansgaard-Oeschger (DO) events of past glacial episodes provide an archetypical example of abrupt climate shifts and are discernible, for example, in oxygen isotope ratios from Greenland ice core records. The physical causes and mechanisms underlying these events are still subjects of ongoing debate. It has previously been hypothesised that DO events may be triggered by bifurcations of physical mechanisms operating at decadal time scales, as indicated by a significant num-
ber of early warning signals (EWS) in the high-frequency variability of records from the North Greenland Ice Core Project (NGRIP). Here, we re-evaluate the presence of EWS by employing indicators based on critical slowing down (CSD) and wavelet analysis and conduct a systematic methodological robustness test. Our findings reveal fewer significant EWS than previous studies, yet their numbers are significant for some of the indicators estimating changes in variability. Additionally, a comparison of different Greenland ice core records also shows significant numbers and consistency for these same EWS esti-
mators preceding a small selection of events in records with high temporal resolution. While those indicators might represent a common climate background, we cannot rule out that signals specific to the different ice core locations are captured. Estimators of correlation times were found to be less consistent and did not provide significant numbers of EWS when considered on their own. Based on these inconclusive results it is not possible to constrain the physical mechanisms underlying the DO events. Instead, our results highlight the complexities and limitations of applying early warning signals to paleoclimate proxy data.

## 1   Introduction

The last glacial period, spanning from approximately 110 to 12 kyears before the year 2000 (b2k), was marked by aperiodic and abrupt climate changes, called Dansgaard-Oeschger (DO) events (Dansgaard et al., 1993, 1982; Johnsen et al., 1992). They are characterised by rapid warming of 5 to 16.5°C (Kindler et al., 2014) over a few decades from colder conditions during Greenland Stadials (GS) to milder ones in Greenland Intersadials (GI), followed by more gradual cooling over centuries or
millennia back to GS (Dansgaard et al., 1982; Johnsen et al., 1992; Rasmussen et al., 2014). DO events were first discovered, and are most evident, in records of oxygen isotope ratios $\delta^{18}O$ from Greenland ice cores (Dansgaard et al., 1993; North Greenland Ice Core Project members et al., 2004), which serve as local temperature proxies. Similar transitions, however, can also be seen in other paleoclimate records including terrestrial archives such as Loess decompositions or speleothems



representing the activity of the tropical monsoon systems(Rousseau et al., 2017; Corrick et al., 2020). While the strongest

expression of DO events was seen in the North Atlantic region (Dansgaard et al., 1982; Johnsen et al., 1992; Dansgaard et al., 1993), they had strong impacts on climate patterns across the globe (e.g Blunier and Brook (2001); Cruz et al. (2005); Wagner et al. (2010); Fohlmeister et al. (2023)).

Despite decades of research, the physical processes behind DO events remain debated. The initially proposed periodicity of approximately 1 470 years suggested that astronomical forces and centennial-scale solar cycles might have influenced these

events (Schulz, 2002), but later studies (Ditlevsen et al., 2007) have indicated that this periodicity might be misleading. Instead, DO variability is often associated with changes in the Atlantic Meridional Overturning Circulation (AMOC), characterized by a weak or shut-off AMOC during GS and strong overturning during GI (see e.g. Lynch-Stieglitz (2017)). However, the specific underlying mechanisms are still not fully understood. Such changes could be driven by external forces (Ganopolski and Rahmstorf, 2001; Knorr and Lohmann, 2003; Zhang et al., 2014, 2017) such as atmospheric carbon dioxide levels (Banderas

et al., 2012; Vettoretti et al., 2022), freshwater discharges from the Laurentide ice sheets (Boers et al., 2022), or volcanic cooling (Lohmann and Svensson, 2022). Nevertheless, shifts in the AMOC and $\delta^{18}$O values in Greenland could also arise from unforced self-oscillation mechanisms (Peltier and Vettoretti, 2014) that are influenced by internal ocean dynamics (Klockmann et al., 2020) and rapid changes in the North Atlantic sea ice (Dokken et al., 2013; Petersen et al., 2013; Boers et al., 2018). The latter is supported by recent advances in comprehensive climate models (e.g. Sakai and Peltier (1997); Vettoretti and Peltier

(2018); Klockmann et al. (2020)), which now depict DO-like events as such oscillations influenced by interactions among sea ice, atmospheric dynamics, and the AMOC (see Malmierca-Vallet et al. (2023) for a review).

DO events provide compelling evidence that abrupt climate transitions over short timescales, relevant for human societies, have occurred in the Earth's past climate system. As such, DO events can be considered archetypes of abrupt climate changes (Boers et al., 2022), which may be caused by crossing system tipping points (TPs). TPs are critical thresholds where a small

perturbation can significantly and non-linearly alter the state or development of a system, often abruptly and/or irreversibly (Lenton et al., 2008), and are a source of growing concern with regards to the potential consequences of ongoing anthropogenic warming. Depending on the mechanisms behind a TP, they can be classified as noise-induced (N-tipping) if a TP is crossed due to internal variations in the system, bifurcation-induced (B-tipping) if tipping occurs by approaching a bifurcation, due to changes in a forcing parameter, where the current state loses stability and the system moves to another stable state, or rate-

induced (R-tipping) if the tipping is not associated with either bifurcation or noise, but is rather caused by rapid changes in the forcing parameter (Ashwin et al., 2012).

Despite a different background climate, similar abrupt transitions may be triggered during current and future warming, where the transition may occur much faster than the change in forcing.

Since the physical mechanisms behind DO events are yet to be clarified, the debate whether they were caused by changes

in an external forcing or through unforced processes, or in other words, the question whether DO events can be considered as examples of N-, or B-tipping is still ongoing. Analyses of dust ($Ca^{2+}$) records from different Greenland ice core sites suggest that DO events might not be purely noise-induced (Lohmann, 2019) and reveal a possible bifurcation structure (Riechers et al., 2023b). Studies of the $\delta^{18}$O record from the North Greenland Ice Core Project (NGRIP, North Greenland Ice Core Project



members et al. (2004)), on the other hand, indicate that these transitions are predominantly noise-induced (Lohmann and
Ditlevsen, 2019) and don't exhibit an underlying bifurcation (Riechers et al., 2023b). Recent conceptual models also propose
different tipping mechanisms for DO events, such as a cascade of tipping points lead by R-tipping of the AMOC due to rapid
sea ice changes (Lohmann et al., 2021), and noise-induced transitions from GS to GI due to fast intermittent anomalies acting
on the sea ice cover (Riechers et al., 2023a).

For systems approaching B-tipping, quantitative indicators that signal the proximity of the system to the TP, so-called
Early Warning Signals (EWS), might potentially be found before the transition. Most common EWS are based on Critical
Slowing Down (CSD): As a system approaches a TP, the stability of the state decreases and its basin of attraction widens.
This is characterized by increasing fluctuation levels and longer correlation times, hence variance $\sigma^2$ and autocorrelation $\alpha_1$
are expected to increase in the observed signal (e.g. Dakos et al. (2008); Ditlevsen and Johnsen (2010)). To capture stability
changes in subcomponents of the system operating on specific timescales, EWS might be constrained to certain frequency bands
of the signal. Accordingly, wavelet-based estimators have been proposed by Rypdal (2016) and further applied in Boers (2018)
for DO events. The scaled-averaged wavelet coefficient $\hat{w}^2$ is used to estimate variance, whilst the local Hurst exponent $\hat{H}$ gives
an estimation of correlation times. In contrast to that, EWS are not expected to occur for purely noise-induced transitions.

While rigorous theory exists for EWS in certain low-dimensional systems (Kuehn, 2011), for instance in analogy with the
Ornstein–Uhlenbeck process (Ditlevsen and Johnsen, 2010), the predictive power of EWS might be limited for complex and
high-dimensional natural systems, such as the Earth's climate (Boers et al., 2022). Even if tipping is due to a bifurcation,
EWS might not be found due to multiple factors, such as the complexity of the underlying system with interactions across
variables that might mask EWS (Morr and Boers, 2024), or an underlying complex bifurcation structure that may not cause
any CSD-based EWS (Morr et al., 2024). Furthermore, the apparent presence of EWS does not automatically imply that a
system approaches a bifurcation since the observed fluctuations may be caused by something else or purely arise by chance
and yield false positives (Boers, 2021). Thus, it is typically assumed that a transition is not entirely noise-induced if EWS are
observed preceding a transition. It can also be helpful to look at multiple EWS indicators simultaneously: Although variance
increases for a system with increasing noise levels that is not approaching a bifurcation, its autocorrelation remains constant
(Ditlevsen and Johnsen, 2010; Smith et al., 2023) . Despite these shortcomings, the presence or absence of EWS for DO events
can give an indication of the underlying tipping mechanisms.

Early warning signals have received a lot of attention in recent years and they are expected to precede potential future
tipping points, e.g., in the polar ice sheets or the AMOC. Climate model studies (van Westen et al., 2024) and analysis of
observational data (Boers, 2021; Ditlevsen and Ditlevsen, 2023) have found EWS for a possible future destabilization of the
AMOC. Nevertheless, the presence of EWS for past abrupt transitions is the only empirical evidence that similar precursors
may be found in observations before future tipping.

While most previous work on EWS for DO events has focused on the abrupt warmings, one study (Mitsui and Boers, 2023)
focused on cooling events from GI to GS during the same time period and found robust CSD-based EWS across $\delta^{18}$O and
dust records from three Greenland ice cores. Several earlier studies have looked for EWS for DO events in $\delta^{18}$O records from
the North Greenland Ice Core Project (NGRIP, North Greenland Ice Core Project members et al. (2004)) with mixed results.



Considering the ensemble average of several DO events, Cimatoribus et al. (2013) find weak but significant CSD-based EWS, whereas Rypdal (2016) later demonstrated that such an average does not yield significant EWS if only the GS preceding DO events are considered. When looking for indications of CSD for individual DO-events across the entire frequency spectrum, Ditlevsen and Johnsen (2010) found no significant EWS preceding any of the 17 events considered there. In contrast to that, Myrvoll-Nilsen et al. (2024) found significant increases for several DO events of the autocorrelation parameter during the preceding GS using a new statistical approach.

Rypdal (2016) limited the search for EWS to high-frequency fluctuations, motivated by the hypothesis that processes operating at time scales shorter than a century are responsible for the rapid, decadal-scale DO transitions. If these are caused by bifurcations, EWS might be detectable in high-frequency bands but masked by low-frequency variability if the entire spectrum is taken into account. To further study such high-frequency fluctuations for individual transitions in the periodicity band between 40 and 60 years, the wavelet-based indicators $\hat{w}^2$ and $\hat{H}$ have been introduced. The author finds some significant EWS for both indicators individually and simultaneously.

A subsequent study (Boers, 2018) re-evaluated the hypothesis of Rypdal (2016) using the raw NGRIP record (North Greenland Ice Core Project members et al., 2004; Gkinis et al., 2014) interpolated to a higher temporal resolution of 5 years instead of the 20 years temporal resolution previously used. There, a significant amount of significant increases in the variance of the 100 year high-pass filtered signal, as well as simultaneous significant increases in variance and autocorrelation is found during GS. Analysis of various frequency bands between 10 and 110 years reveals most wavelet-based EWS in a scale range of 10 to 50 years, where a significant amount of significant EWS is found for $\hat{w}^2$, $\hat{H}$ and both occurring simultaneously. These results suggested that DO events might have occurred due to B- rather than N-tipping.

Previous EWS analyses for DO warming transitions have all been conducted on $\delta^{18}O$ records from the NGRIP ice core in various temporal resolutions but other available $\delta^{18}O$ records from other ice cores (Fig. 1), that clearly exhibit the same DO events (Rasmussen et al., 2014) as it can be seen in Fig. 2, have not been taken into account. This raises the question whether the high-frequency $\delta^{18}O$ variability from different Greenland ice core records is comparable during GS before transitions and whether similar EWS can be found across different records and temporal resolutions.

Here we re-evaluate the results from Boers (2018) across multiple Greenland ice cores (Sect. 3.3 and 3.6). We conduct a systematic comparison of EWS during GS before DO events for a total of six $\delta^{18}O$ records from four ice core sites in three different temporal resolutions (see Fig. 1, Fig. 2 and Sect. 2.1) to assess whether the observed high-frequency fluctuations prior to DO events 1–16 (counting from younger to older events, see Fig. 2, Svensson et al. (2008)) and the Younger Dryas-Preboreal transition (YD/PB, at approx. 11 700 years b2k, Svensson et al. (2008)) stem from a common climate background or could have been caused by other factors. The early warning indicators considered, variance $\sigma^2$, lag-1 autocorrelation $\alpha_1$, wavelet fluctuation level $\hat{w}^2$, and Hurst exponent $\hat{H}$, are the same as used by Boers (2018), where we apply some modifications to the methods presented there (see Sect. 2.5). Moreover, we evaluate the robustness of EWS on these methodological changes, i.e. different choices in significance testing, EWS estimation, and data preprocessing for the NGRIP record with 5-year temporal resolution (Sect. 3.1 and 3.4). To circumvent potential interpolation effects, we further conduct a similar study on the raw NGRIP record applying an approach adapted specifically for the analysis of irregularly sampled time series (Sect. 3.2 and 3.5).







**Figure 1.** Map of Greenland with the locations of the deep ice core drilling sites GRIP (72.58° N, 37.64° W), GISP2 (72.58° N, 38.48° W), NGRIP (75.10° N, 42.32° W), and NEEM (77.45° N, 51.06° W) marked in red.







**Figure 2.** Greenland $\delta^{18}$O proxy records from NGRIP in 5- (a), 10- (b), and 20-year (d) resolution, NEEM in 10-year resolution (c), GRIP (e), and GISP2 (f) in 20-year resolution. Time series during GS studied here are shown in blue, their onsets are marked with blue vertical lines. DO events and the YD/PB transition are marked by the red vertical lines and define the onsets of GI, drawn in red.



## 2 Methods

### 2.1 Data and preprocessing

We consider all available $\delta^{18}$O records from Greenland ice cores between 59 920 yr b2k and 10 295 yr b2k on the associated annual-layer counted Greenland Ice-Core Chronology 2005 (GICC05, Rasmussen et al. (2006); Andersen et al. (2006); Svensson et al. (2006)) with a temporal resolution of at least 20 years.

For the estimation of CSD-based EWS $\sigma^2$ and $\alpha_1$, we use the 100-year high-pass filtered data of the normalised time series. This is achieved by applying a Chebychev Type-I high-pass filter with cutoff at 100 years.

#### 2.1.1 Ice core data in 20-year resolution

The three $\delta^{18}$O records from NGRIP (North Greenland Ice Core Project members et al., 2004), the Greenland Ice Core Project (GRIP, Johnsen et al. (1997)) and the Greenland Ice Sheet Project Two (GISP2, Grootes and Stuiver (1997); Stuiver and Grootes (2000)) have been synchronised and resampled at 20-year resolution (Rasmussen et al., 2014; Seierstad et al., 2014). The data are available as step data and we associate each $\delta^{18}$O value with its later age (i.e. $x(t_i) \in \{x_{i-1}, x_i\}$, where we use $x(t_i) \rightarrow x_i$ for all ages $t_i$ and $\delta^{18}$O values $x_i$). In the GISP2 record there are $n = 24$ missing $\delta^{18}$O values throughout the entire time interval, of which $n_{GS} = 12$ occur during GS: $n_1 = 4$ in the GS before DO-1, $n_2 = 2$ prior to DO-2, $n_4 = 3$ preceding DO-4, and $n_5 = n_6 = n_7 = 1$ before DO-5, DO-6, and DO-7, respectively. We replace these missing data points by random values from a normal distribution of a 120-year range around the value within the same GS or GI, respectively.

#### 2.1.2 Ice core data in 10- and 5-year resolution

The North Greenland Eemian Ice Drilling (NEEM, Gkinis et al. (2020)) ice core provides a $\delta^{18}$O record sampled in 5 cm depth resolution and associated ages are available in the GICC05 time scale, yielding an average time step of 4.18 years, where only 0.09% of temporal sampling steps are > 10 years (Fig. A2). To obtain equal spacing in time, we interpolate the raw NEEM data to a regular 10-year resolution. The raw NGRIP $\delta^{18}$O record in 5 cm depth steps (North Greenland Ice Core Project members et al., 2004) provides an average time step of 2.43 years, where all sampling steps are < 10 years and only 0.46% of temporal sampling steps are > 5 years (Fig. A1). To be able to compare EWS of the NGRIP record in different time resolutions, we interpolate to regular 5- and 10-year steps, respectively. To do so, we first interpolate the raw signal to yearly time steps using cubic splines. After applying a Chebychev Type-I low-pass filter with cutoff at half the desired sampling frequency to avoid aliasing effects, we resample the records every 5 and 10 years, respectively. Interpolating the raw signals directly to the desired temporal resolutions without using a low-pass filter yields different, yet similar results for the presence of EWS. These are shown in the Supplementary Sect. S2.




## 2.2 EWS calculation

We search for EWS during the GS prior to DO events 1–16 and the PB/YD transition, where we use the same definitions of GS and GI as Boers (2018), given there in Supplementary Table 1.

### 2.2.1 CSD indicators

Variance $\sigma^2$ and the lag-1 autocorrelation coefficient $\alpha_1$ are calculated in moving windows of 200 years width, shifted over the 100-year high-pass filtered, regularly spaced $\delta^{18}$O time series during GS. Windows with less than 200 years of data are ignored to ensure that the transition itself is not taken into account.

For the irregularly-sampled NGRIP record, we estimate indicators of the band-filtered signal, obtained from the amplitude scalogram (see Lenoir and Crucifix (2018a) for details) for time scales $s \in [\min(s), \min(\max(s), 100)]$ during GS preceding DO events. Variance is calculated as for the regularly spaced data. We calculate the approximated autocorrelation coefficient $\hat{\alpha}_1$ in 200-year moving windows during GS from the estimated persistence time $\tau$ as described by Mudelsee (2002) as $\hat{\alpha}_1 = e^{-\bar{d}/\tau}$, where $\bar{d}$ is the mean temporal spacing.

### 2.2.2 Wavelet-based indicators

To obtain the wavelet-based indicators, we estimate the wavelet power spectra $|W_t(s)|^2$ of the $\delta^{18}$O time series separately for each GS preceding transitions and exclude all times $t$ for which the wavelet power lies within the cone of influence (COI, the region in the wavelet spectrum, where edge effects become important) to avoid uncertain estimations of the spectrum and any influence of the transition itself. We choose the Paul wavelet basis (of order 4), as done by Rypdal (2016) and Boers (2018). In order to compare the results to indicators obtained from the irregularly sampled NGRIP data, we also apply the Morlet wavelet basis (with parameter $\omega_0 = 6$) to the NGRIP time series with 5-year resolution. A detailed introduction to wavelets can be found in Torrence and Compo (1998).

The scale-averaged wavelet coefficient $\hat{w}^2_{s_1,s_2}$ yields a time series of the average variance in a periodicity band between scales $s_1$ and $s_2$ and is given by the weighted average of the wavelet power spectrum as

$$\hat{w}^2_{s_1,s_2} = \frac{\delta j \delta t}{C_\delta} \sum_{j=j_1}^{j_2} \frac{|W_t(s_j)|^2}{s_j}, \tag{1}$$

where we use the reconstruction factor $C_\delta = 1.132$ when using the Paul wavelet basis, and $C_\delta = 0.776$ for Morlet (Torrence and Compo, 1998). The scale resolution is set to $\delta j = 0.1$ and the temporal resolution $\delta t$ is chosen to be the temporal resolution of the data.

To compute the time series of the local Hurst exponent $\hat{H}$ as an estimate of correlation times, we use the following scaling of the variance $V_W(s)$ of the wavelet transform $W_t(s)$:

$$V_W(s) = \frac{|W_t(s)|^2}{s} \sim s^{2\hat{H}-1}. \tag{2}$$



For a more detailed description, see Rypdal (2016). Wavelet-based techniques and Hurst analysis for scaling processes are thoroughly summarised by Malamud and Turcotte (1999). Consequently, we get

$$\hat{H}_{s_1,s_2} = \frac{a_{s_1,s_2} + 1}{2}, \tag{3}$$

where $a_{s_1,s_2}$ denotes the slope of a linear fit between $\log(s)$ and $\log(|W_t(s)|^2/s)$ for scales $s_1 \leq s \leq s_2$ at each time $t$. We consider scale ranges $(s_1, s_2)$ where $s_1 < s_2$ with $s_1 \in \{10, 20, \ldots, 100\}$ and $s_2 \in \{20, 30, \ldots, 110\}$ for the records with 5- and 10-year resolution. For the records sampled every 20 years, we choose $s_1 \in \{20, 40, 60, 80\}$ and $s_2 \in \{40, 60, 80, 100\}$. For simplicity, we denote $\hat{w}^2 := \hat{w}^2_{s_1,s_2}$ and $\hat{H} := \hat{H}_{s_1,s_2}$ when the context clearly specifies the range of scales between $s_1$ and $s_2$ years.

We compute the (irregularly sampled) wavelet power spectra of the raw NGRIP $\delta^{18}$O record as described by Lenoir and Crucifix (2018a) and implemented in the WAVEPAL[1] package for (time-)frequency analysis of irregularly sampled time series, based on Lenoir and Crucifix (2018a) and Lenoir and Crucifix (2018b). This approach uses the Morlet wavelet basis, where we choose the parameter $\omega_0 = 6$. The indicators $\hat{w}^2$ and $\hat{H}$ are then calculated as described above, using Eq. (1), (2), and (3).

### 2.3 Testing for significant trends

To test for significant positive trends of the indicator time series, we create $n = 10,000$ truncated Fourier transform (TFTS) surrogates (Nakamura et al., 2006) for each (high-pass filtered) $\delta^{18}$O record during every GS by randomising the phases in Fourier space, but keeping the lowest 5% of frequencies unchanged to account for possible trends in the signal. This choice of surrogates allows to handle data with irregular fluctuations superimposed over long term trends, without the need for manual detrending of the signal. Thus, we test against the null hypothesis that the irregular fluctuations of the signal are generated by a stationary linear system (Nakamura et al., 2006). Similar to Fourier surrogates, where all Fourier phases are shuffled, TFTS surrogates preserve the variance and autocorrelation function of our original time series (see Supplementary Fig. S1 and S2).

For significance testing on the irregularly sampled $\delta^{18}$O NGRIP record, TFTS surrogates cannot be used since the Fourier transform cannot be computed for such data. Instead, we apply a similar approach and shuffle all but the lowest 5% of frequencies of the amplitude scalogram of the (band-filtered) $\delta^{18}$O data during GS before reconstructing the signal to construct surrogates. Due to the higher computational time, only $n = 1,000$ surrogates are considered in this case.

EWS estimation is performed for the resulting surrogates as for the original data during GS and we calculate the linear trends ($a_0$) of the EWS indicators of the original time series and their surrogates ($a_s$). We consider an increase in the indicators to be significant if its trend is positive, i.e. $a_0 > 0$, and if the right-sided $p$-value $p = P(a_s \geq a_0) < 0.05$. Examples of the resulting null-model distributions of linear trends are depicted in Supplementary Fig. S3–S6.

### 2.4 Expected number of spurious significant EWS

With our chosen method of significance testing, spurious significant EWS of a linear stochastic process are expected at a probability of 5% by definition. Assuming that the occurrences of significant EWS for the 17 transitions are independent,

---

[1]https://github.com/guillaumelenoir/WAVEPAL



the number of false positives within one $\delta^{18}$O record should follow a binomial distribution $B(n,p)$ with $n = 17$ trials and success probability $p = 0.05$. For $x \sim B(17, 0.05)$, it is $P(x \leq 2) \approx 0.9497 < 0.95$ and $P(x \leq 3) \approx 0.9912 > 0.95$. Thus, at a confidence level of 95%, we expect at most two events to show spurious significant early warning, and observing three

significant EWS is statistically significant.

To verify this analytic result numerically for the NGRIP record in 5-year resolution, we generate $m = 2,000$ TFTS surrogates ($m = 1,000$ for the local Hurst exponent $\hat{H}$ due to computational reasons) of the entire time series containing the 17 transitions. For each of these surrogates, we place 17 GS of original length randomly and calculate the number of significant EWS for $\sigma^2$, $\alpha_1$, and the wavelet-based estimators $\hat{w}^2$ and $\hat{H}$ in the scale band between 10 and 50 years using $1,000$ surrogates for each

event. The resulting distributions of expected spurious EWS can be seen in Fig. 3(a),(b) and Fig. A3(a),(b). They show a close resemblance to the binomial distribution $B(17, 0.05)$ for all indicators. The numerical results indicate that observing two significant increases in the autocorrelation $\alpha_1$ and the scale-averaged wavelet-coefficient $\hat{w}^2$ is statistically significant, while they confirm this number to be three for $\sigma^2$, and $\hat{H}$ at 95% confidence. These differences in the significance thresholds despite the similarity of distributions can be explained by the discrete nature of the distributions.

For a linear stochastic process, we would expect increases in variability and correlation times to be independent. Hence, the number of spurious significant increases in two indicators, $\sigma^2$ and $\alpha_1$, or $\hat{w}^2$ and $\hat{H}$ simultaneously, is expected to follow the binomial distribution $B(n, p^2)$. At 95% confidence, no such simultaneous increase is expected (Fig. 3(c) and A3(c)).





**Figure 3.** Null-model distributions for the number of significant EWS in $\sigma^2$ (a), $\alpha_1$ (b), and both CSD-indicators simultaneously (c) for the NGRIP $\delta^{18}$O record with 5-year resolution.

## 2.5 Overview of method modifications

While our approach to data processing, EWS calculation and significance testing described above is based on the work by
Boers (2018), some details differ from those applied there. Table 1 provides an overview of our modifications. We follow steps
1, 2a, and 3 for the CSD-based indicators, and steps 1, 2b, and 3 for their wavelet-based counterparts.



| Step | Method used by Boers (2018) | Modification |
|------|------------------------------|--------------|
| 1 | **SIGNIFICANCE TESTING** | |
| 1.1 | Indicators calculated on entire time period | Indicators calculated in GS only |
| 1.1 | Surrogates of indicators | Surrogates of data |
| 1.1 | Fourier surrogates | TFTS surrogates |
| 2a | **CSD-BASED EWS ESTIMATION** | |
| 2.1a | 800-year low-pass filtered indicators | No filtering of indicators |
| 2.2a | EWS in GS until 200 years before transition | EWS in entire GS, only windows with 200 years of data considered |
| 2b | **WAVELET-BASED EWS ESTIMATION** | |
| 2.1b | 800-year low-pass filtered indicators | No filtering of indicators |
| 2.2b | 200-year average of $\hat{w}^2$ and $\hat{H}$ | Using indicators directly |
| 2.3b | EWS in GS until 200 years before transition | EWS in entire GS, exclusion of COI |
| 3 | **DATA PREPROCESSING** | |
| 3.1 | preprocessing in Python 2.7 | preprocessing in Julia 1.10 |
| 3.2 | ages in raw data rounded to 1/10 years | exact ages in raw data |

**Table 1. Overview of method modifications compared to Boers (2018).** Modifications to the methods used there are applied sequentially to the significance testing (Step 1), EWS estimation (Steps 2a and 2b for CSD- and wavelet-based indicators, respectively), and data processing (Step 3).

### 2.5.1 Significance testing

Rather than constructing surrogates by randomising the phases of the detrended indicator time series, we use the $\delta^{18}$O signal itself and keep the lowest 5% of frequencies unchanged to account for possible trends in the data, without detrending manually.
In order to construct surrogates of the data whilst still following the same procedure for surrogates and the $\delta^{18}$O record, we consider the indicator time series during GS individually. This differs from the approach by Boers (2018), where indicators were calculated over the entire time period and slices during GS were considered to search for EWS.

### 2.5.2 EWS estimation

In contrast to Boers (2018), we do not apply a Chebyshev Type-I low-pass filter with cutoff at 800 years to extract millennial
scale variability of the high-frequency indicator time series, but rather look for EWS in the indicator time series directly. We further note that such a filter does not yield an effect on the relatively short (35 – 8 215 years; avg. 1 588 years) time series during GS considered here.



Instead of searching for significant increases of variance and autocorrelation in the GS until 200 years before each transition using centered 200-year moving windows, we consider the entire GS but discard windows which contain less than 200 years
of data.

To reap the advantage that using wavelet methods does not require moving time windows, we do not apply a 200-year average to $\hat{w}^2$ in Eq. (1). Similarly, we calculate the Hurst exponent $\hat{H}$ for each time $t$ directly without applying a moving 200-year average to $|W_t(s)|^2/s$ in Eq. (2) as done by Rypdal (2016) and Boers (2018). Furthermore, we don't restrict the search for wavelet-based EWS to the GS until 200 years prior to events to include potential influences of the transitions themselves.
Instead, the entire GS is considered and any time points within the COI are discarded. Additionally, we consider the wavelet power spectra of the regularly sampled $\delta^{18}O$ time series directly without normalisation.

### 2.5.3 Data preprocessing

Even though we follow the same steps in data preprocessing as Boers (2018), small differences between the $\delta^{18}O$ records and thus the indicator time series arise. This is due to numerical differences and different implementations of e.g. the low- and
high-pass filters between Python 2.7 used there and Julia 1.10 used here. Moreover, we analyse the publicly available NGRIP record, that differs slightly from the one used by Boers (2018), where the ages were rounded to one-tenth of a year.

## 3 Results

### 3.1 CSD-based early warning signals in the NGRIP record with 5-year resolution

When searching for EWS, many methodological choices have to be made. Here, we systematically test the robustness of early
warning signals to a variety of such choices. To do so, we analyse the methods of Boers (2018) and sequentially evaluate modifications in the significance testing, EWS calculation, and data preprocessing for the high-frequency variability of the NGRIP record, following steps 1, 2a and 3 in Table 1 and described in Sect. 2.5. A full overview of the influences of the individual modifications following steps 1.1, 2.1a, 2.2a, 3.1, and 3.2 is shown in Supplementary Fig. S12.

While attempting to recreate the results of Boers (2018), we find significant EWS for 11 out of 17 transitions in the variance
$\sigma^2$, seven in the autocorrelation $\alpha_1$, and five for both indicators simultaneously (Fig. 4(a,b)). This differs from the results of Boers (2018) who show an additional event with a significant increase in variance ($n_{\sigma^2} = 12$, $n_{\alpha_1} = 7$, and $n_{both} = 6$). The additional EWS in $\sigma^2$ stems from an erroneous calculation, where the time series of the scale-averaged wavelet coefficient $\hat{w}^2$ was considered instead of the variance $\sigma^2$.

As a first robustness test, we modify how surrogates are obtained for significance testing and construct surrogates of the data
during GS prior to transitions, instead of the indicator time series. This decreases the number of significant EWS from 11 to 4 in $\sigma^2$, and from 7 to 2 in $\alpha_1$. Only one event (DO-12) shows a simultaneous significant increase in both $\sigma^2$ and $\alpha_1$. (Fig. 4(c,d)). We note that the resulting indicator time series differ slightly because applying a 800-year low-pass filter, as done by Boers (2018) and in Fig. 4(a,b), doesn't yield the same effect when applied to the GS rather than the entire time period.





Next, when modifying how $\sigma^2$ and $\alpha_1$ are calculated, the previously significant EWS remain. For the variance, one event

(DO-8) that shows a significant increase with the initial significance testing, but not the modified one, now displays early warning. As in the previous step, only one event is preceded by precursors in both variance in autocorrelation, i.e. $n_{\sigma^2} = 5$ (prior to DO-1, 4, 6, 8, and 12), $n_{\alpha_1} = 2$ (prior to DO-10 and 12), and $n_{\text{both}} = 1$ (prior to DO-12) (Fig. 4(e,f)).

Finally, we change how the $\delta^{18}\text{O}$ record is preprocessed. This does not yield any changes to the CSD-based early warning signal of the high-frequency variability of the NGRIP record (Fig. 4(g,h)).

According to both, the binomial and numerically constructed null-distributions for spuriously appearing early warning signals (Fig. 3 and Sect. 2.4), observing five significant increases in $\sigma^2$ is statistically significant at 95% confidence. This is also the case for the simultaneous warning from variance and autocorrelation for DO-12 Though, observing two significant EWS in $\alpha_1$ is only significant with respect to the analytical, but not the numerical null-distribution.





**Figure 4.** Early warning signals in the variance and autocorrelation of the 5-year interpolated and 100-year high-pass filtered NGRIP $\delta^{18}$O record with sequential method modifications. (a) Time series of the variance (black) during GS calculated using the methods described by Boers (2018). (b) Same as (a) but for the lag-1 autocorrelation coefficient. (c-d) Same as (a-b) but with modified significance testing. (e-f) Same as (c-d) but with modified indicator calculation.(g-h) Same as (e-f) but with modified data preprocessing.DO events and the YD/PB transition are marked by the red vertical lines. Linear trends of the indicators are shown by red (blue) lines and the corresponding pale shading of the GS period if the trend is positive (negative). Significant linear increases are indicated by a dark red shading of the GS preceding transitions. The number of significant increases in $\sigma^2$, $\alpha_1$ and both indicators simultaneously are denoted by $n_{\sigma^2}$, $n_{\alpha_1}$ and $n_{\text{both}}$, respectively.





## 3.2 CSD-based early warning signals in the NGRIP record with irregular temporal resolution

Using spectral methods adapted to irregular time sampling as described in Sect. 2 for the raw, irregularly sampled NGRIP $\delta^{18}$O record, we find four significant EWS in $\sigma^2$ (prior to DO-1, 4, 6, and 12), and three in $\hat{\alpha}_1$ (for DO-6, 10, and 12), where two events (DO-6 and 12) show synchronous significant increases in both indicators (Fig. 5).

While all significant variance increases in the raw time series are also found in the interpolated record with even time sampling, there is one event (DO-8) that is not preceded by an early warning here (Fig. 4(g), 5(a)). Two of the three GS
(prior to DO-10 and 12) displaying significant $\hat{\alpha}_1$ increases here show significant increases in $\alpha_1$ of their regularly sampled counterparts. In both cases, DO-12 is preceded by significant EWS in both CSD-estimators, and analysis of the irregularly sampled raw record reveals another simultaneous warning for DO-6 (Fig. 4(h), 5(b)).

Considering the binomial null distributions for false positives, the observed number of significant increases in the variance, the autocorrelation, and both CSD-indicators simultaneously, is statistically significant at 95% confidence.

**EWS in 100-year high-pass filtered NGRIP record**

Irregular temporal resolution : $n_{\sigma^2} = 4$, $n_{\hat{\alpha}_1} = 3$, $n_{both} = 2$

**Figure 5.** Early warning signals in the variance and autocorrelation of the raw, irregularly sampled, 100-year high-pass filtered NGRIP $\delta^{18}$O record. (a) Time series of the variance (black) during GS (b) Same as (a) but for the autocorrelation coefficient. Line colours and shadings are applied in the same way as in Fig. 4.

## 3.3 CSD-based early warning signals across ice core records


EWS in the variance and lag-1 autocorrelation coefficient of the the various $\delta^{18}$O records from Greenland ice cores are shown in Fig. 6. There, we can see that only NGRIP with 5-year sampling steps shows a significant EWS for $\sigma^2$ and $\alpha_1$ simultaneously (DO-12).

The number of significant variance increases ranges from zero (GRIP, 20-year resolution) to five (NGRIP, 5-year resolution).
For the autocorrelation, this number ranges from zero to four, but in this case NEEM in 10-year resolution and GRIP in 20-year resolution display the fewest, whereas GISP2 in 20-year resolution the most EWS. The numbers of significant increases in the variance are statistically significant for the NGRIP record with 5- and 10-year resolution, as well as for NEEM. For the



autocorrelation, only GISP2 in 20-year resolution displays a significant number of significant EWS at 95% confidence.These significance thresholds are taken with respect to the binomial distribution $B(17, 0.5)$ to be able to compare the different records.

We find three common significant increases in variance across the different records for DO-1 for the signal from NGRIP in 5- and 20-year resolutions, as well as NEEM. This event is also preceded by a common increase in the autocorrelation in the 10-year NGRIP record and the 20-year GISP2 record. Furthermore, DO-12 and DO-6 display significant increases in $\sigma^2$ for all the high-resolution records (NGRIP 5- & 10-year, NEEM 10-year sampling). One additional event, DO-8, is preceded by a common significant EWS of the variance for the NGRIP record in 5- and 20-year resolutions.Another common significant

increase in the autocorrelation can be seen for DO-10 in NGRIP with 5-year resolution and GISP2.

Regarding the EWS of $\sigma_2$ in the NGRIP record across different temporal resolutions, we note that the 5-year and 10-year resolution signals share two common events (DO-12 and DO-6) with preceding EWS. The 5- and 20-year sampled records share the two present for the latter (DO-1 and DO-8), and the ones with 10- and 20-year time steps have no significant variance increase in common. There is further no common significant increase in the autocorrelation across different temporal

resolutions for the NGRIP record.

While we seem to find more variance increases for the high-resolution records (five for 5-year, three for 10-year and zero, one and two for 20-year), there is hence no such apparent trend for the autocorrelation.

Comparing records with the same temporal resolution, we find two common significant EWS in $\sigma^2$ (DO-12 and DO-6) for NGRIP and NEEM, sampled every 10 years, and none for the time series with 20-year time steps.

For a comparison of common EWS across records, see also Supplementary Fig. S7.





**Figure 6.** Early warning signals in the variance and autocorrelation of various 100-year high-pass filtered Greenland $\delta^{18}$O records. (a) Time series of the variance (black) during GS of the 5-year NGRIP record. (b) Same as (a) but for the lag-1 autocorrelation coefficient. (c-d) Same as (a-b) but for the 10-year NGRIP record. (e-f) Same as (a-b) but for the 10-year NEEM record. (g-h) Same as (a-b) but for the 20-year NGRIP record. (i-j) Same as (a-b) but for the 20-year GRIP record. (k-l) Same as (a-b) but for the 20-year GISP2 record. Line colours and shadings are applied in the same way as in Fig. 4.





### 3.4 Wavelet-based early warning signals in the NGRIP record with 5-year resolution

As an alternative approach to the CSD-EWS $\sigma^2$ and $\alpha_1$, we also look for significant increases of the scale-averaged wavelet coefficient $\hat{w}^2$ and the local Hurst exponent $\hat{H}$ preceding DO events. To be able to compare our results with those obtained by Boers (2018), we focus on the 10–50 year periodicity band. Besides the methodological modifications presented for the CSD

indicators above, we include changes specific to these wavelet-based EWS and apply them to the NGRIP $\delta^{18}$O record with 5-year time steps. We proceed in the same manner as before and follow steps 1, 2b and 3 in Table 1 (see Sect. 2.5 for details). The subsequent results are depicted in Fig. 7. Supplementary Figure S13 provides a more detailed synopsis following steps 1.1, 2.1b, 2.2b, 2.3b, 3.1, and 3.2.

When applying the methods described by Boers (2018), we find the same significant EWS, i.e. 12 significant increases in

$\hat{w}^2$, 8 in $\hat{H}$, and 7 in both indicators simultaneously (Fig. 7(a,b)).

As for the CSD-based indicators, our modifications in significance testing result in fewer significant EWS in both indicators with $n_{\hat{w}^2} = 4$ (for DO-1, 4, 6, and 12), $n_{\hat{H}} = 2$, and $n_{\text{both}} = 2$ (for DO-1 and 12) (Fig. 7(c,d)).

Further modifications to the EWS estimation also lead to four significant increases in $\hat{w}^2$ (preceding DO-1, 6, 7 and 12). While two of them (for DO-1 and 6) were significant in the previous step, one increase of $\hat{w}^2$ lost its significance, and another

one (prior to DO-7) became significant again (Fig. 7(c,e)). For $\hat{H}$, one increase loses significance, resulting in only one event (DO-1) with a significant EWS in the local Hurst exponent, as well as both wavelet-based stability estimators simultaneously (Fig. 7(e,f)).

Additional alterations in data preprocessing do not yield any further changes for the wavelet-based early warning indicators (Fig. 7(g,h)).

We observe that three of the four events (DO-1, 6, and 12) displaying significant EWS in $\hat{w}^2$ also show significant increases in $\sigma^2$, whereas significant EWS in $\alpha_1$ and $\hat{H}$ do not coincide (Fig. 4(g,h), 7(g,h)).

As for variance and autocorrelation, we construct a null-distribution for the number of false EWS both analytically and numerically (see Sect. 2.4, Fig. A3).

The number of significant EWS in $\hat{w}^2$ is statistically significant at 95% with respect to both of them, whereas this number

is not significant for the local Hurst exponent considering either null distribution. Considering the binomial distribution for the number of synchronous increases of the indicators, we find that observing one simultaneous EWS in $\hat{w}^2$ and $\hat{H}$ is statistically significant with 95% confidence.

Even though the numbers of significant EWS in the high-frequency variability of the NGRIP $\delta^{18}$O record, $n_{\sigma^2} = 5$ and $n_{\hat{w}^2} = 4$, could potentially be seen as evidence for a destabilisation of the system, those for the correlation times, $n_{\alpha_1} = 2$ and

$n_{\hat{H}} = 1$, do not indicate a consistent widening of the basin of attraction associated with mechanisms operating on decadal time scales, across the series of DO events.

Furthermore, we note that while both CSD- and wavelet-based indicators show a statistically significant simultaneous significant increase in variability and correlation times, these do not occur for the same transitions (DO-12 for the CSD indicators (Fig. 4(g,h)), and DO-1 for the wavelet-based ones (7(g,h)), respectively).





**Figure 7.** Early warning signals in the wavelet-based indicators confined to the 10–50-year periodicity band of the 5-year interpolated NGRIP $\delta^{18}O$ record with sequential method modifications. (a) Time series of the scale-averaged wavelet coefficient (black) during GS, calculated using the methods described by Boers (2018). (b) Same as (a) but for the local Hurst exponent. (c-d) Same as (a-b) but with modified significance testing. (e-f) Same as (c-d) but with modified estimator calculation.(g-h) Same as (e-f) but with modified data preprocessing.Line colours and shadings are applied in the same way as in Fig. 4.





## 3.5 Wavelet-based early warning signals in the NGRIP record with irregular temporal resolution

For the raw NGRIP $\delta^{18}$O record with variable time steps, the classical wavelet methods used for regularly sampled time series cannot be applied. Instead, we make use of the adapted methods introduced by Lenoir and Crucifix (2018a) and described in Sect. 2.2.2. A technical difference between this approach and the one we used for the regularly sampled records is the choice of the Morlet wavelet as the mother wavelet instead of the Paul wavelet, that we used throughout our analysis so far. Thus, to compare EWS between the raw and interpolated data, the analysis of the interpolated time series is repeated using the Morlet wave basis here.

When searching for wavelet-based EWS in the raw $\delta^{18}$O NGRIP record, we find four significant EWS in the scale-averaged wavelet coefficient $\hat{w}^2$ (for the YD/PB transition, DO-1, 4, and 6) and two in the local Hurst exponent $\hat{H}$ (for DO-12 and 5). None of the 17 events show simultaneous increases in both indicators in the 10–50 year periodicity band (Fig. 8(c,d)). Looking at the interpolated time series, we observe two EWS in $\hat{w}^2$ (prior to DO-6 and 12), and three in $\hat{H}$ (prior to DO-1, 10, and 12), of which one precedes the same event (DO-12) in both indicators (Fig. 8(a,b)). Comparing the two versions of the NGRIP record, we see that DO-12 is preceded by significant increases in $\hat{H}$ for both of them. Further, they share one common significant increase in $\hat{w}^2$ prior to DO-6 (Fig. 8).

The raw record displays a significant number of significant increases in the scale-averaged wavelet coefficient at 95% confidence with respect to a binomial null-distribution. The number of significant EWS in the local Hurst exponent might be spurious for either record. Nonetheless, the occurrence of a simultaneous significant increase in both indicators prior to DO-12 in the interpolated record is statistically significant.

While using the Morlet mother wavelet yields two significant increases less in $\hat{w}^2$ compared to their estimation using the Paul wavelet, we find two additional significant EWS in $\hat{H}$. Either choice of wavelet function yields one event with a simultaneous increase in both indicators. Nevertheless, these occur for different events: DO-1 using Paul (Fig. 7(g,h)) and DO-12 using Morlet (Fig. 8(a,b)).





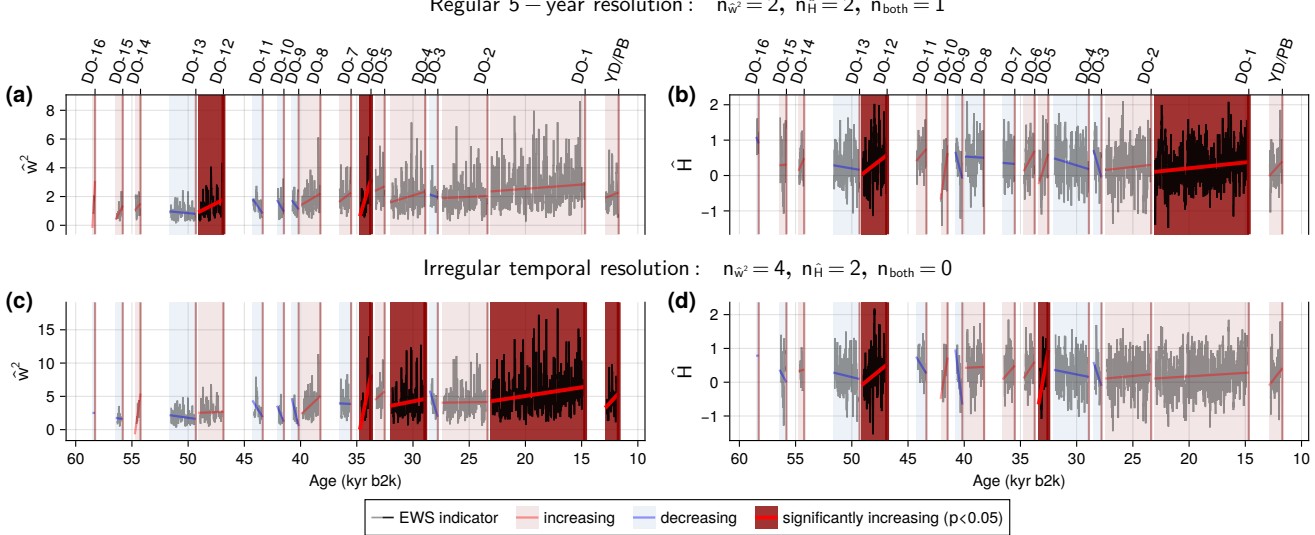

**Figure 8.** Early warning signals in the wavelet-based indicators confined to the 10–50-year periodicity band of the 5-year interpolated and raw NGRIP $\delta^{18}$O records. (a) Time series of the scale-averaged wavelet coefficient (black) during GS of the 5-year interpolated NGRIP record using the Morlet wavelet function. (b) Same as (a) but for the local Hurst exponent. (c-d) Same as (a-b) but for the raw, irregularly sampled NGRIP record. Line colours and shadings are applied in the same way as in Fig. 4.

## 3.6 Wavelet-based early warning signals across ice core records

To be able to compare EWS between the various ice core records with different temporal resolutions, ranging from 5 to 20 years, we focus on $\hat{w}^2$ and $\hat{H}$ in the 20–60 year frequency band instead of the 10–50 year one considered before. The results

are depicted in Fig. 9 which reveals that two of the records, NEEM in 10- and NGRIP in 20-year resolution, show significant EWS in both wavelet-based indicators simultaneously (for DO-1 and DO-2, respectively). These are statistically significant results at the 95% confidence level. Nevertheless, we note that DO-2 is not preceded by any significant EWS in any other record considered here.

For $\hat{w}^2$, the number of significant increases ranges from one (GRIP and GISP2 with 20-year resolution) to three (NGRIP

with 5-year sampling and both 10-year resolution records) and for $\hat{H}$ from zero (NGRIP with 5- and 10-year resolution) to two (GISP2). None of these numbers is statistically significant at 95%.

Similarly to the CSD-based EWS, none of the 17 events is preceded by common significant EWS across all records. Nevertheless, DO-1 is anticipated by significantly increasing $\hat{w}^2$ in most records (except for GRIP), and significant EWS in $\hat{H}$ in the NGRIP record interpolated to 10-year time steps. Two more events, DO-6 and 12, display significant EWS in $\hat{w}^2$ in the



higher resolution records, i.e. NGRIP with 5- and 10-year resolution and NEEM. The same can be seen for the variance (Fig. 6(a,c,e)). There is no common increase of the local Hust exponent across any of the records.

   While the NGRIP record only shows one significant EWS in the local Hurst exponent if sampled every 20 years, we find a common significant increase in the scale-averaged wavelet coefficient for DO-1 across all temporal resolutions, and DO-6 and DO-12 are preceded by significant increases for 5- and 10-year sampling steps.

Looking at time series with the same temporal resolution, we see that all three significant increases in $\hat{w}^2$ prior to DO-1, 6, and 12 are common across the 10-year resolution records. DO-1 also has a common significant EWS in the scale-averaged wavelet coefficient for NGRIP and GISP2, but not across all 20-year records.

   When comparing these results to the ones for the CSD-based early warning signals, we find that the indicators for high-frequency variability $\sigma^2$ and $\hat{w}^2$ have common increases during three GS (prior to DO-1, 6, and 12) for the NGRIP record

sampled every 5 years and NEEM, two (prior to DO-6 and 12) for NGRIP at 10-year and one (preceding DO-1) for NGRIP with 20-year resolution. Regarding the estimators of correlation time, $\alpha_1$ and $\hat{H}$, only the $\delta^{18}$O record obtained from GISP2 shows a common increase in both of them, preceding DO-10 (Fig. 6, 9).

   An overview of wavelet-based EWS in the different frequency bands $(s_1, s_2)$ relevant for all of the considered records can be found in Fig. A4. There, we see that there is no such band with a common significant indicator increase in all of the records, but

common significant increases of $\hat{w}^2$ for DO-1 are found in all records but GISP2 in the (20,60), (20,80), and (20,100) year scale ranges. Furthermore, DO-6 and DO-12 are preceded by common significant EWS of the higher resolution records for a range of frequency bands with the lower bound $s_1 \leq 20$ and $s_1 \leq 40$, respectively. Regarding the Hurst exponent, fewer common significant EWS are found, where the NGRIP record displays significant increases in all resolutions for DO-2 when the (20,80) and (40,80) year bands are considered. We only observe one common significant increase in both indicators simultaneously

prior to DO-1 for NGRIP in 5- and 10-year resolution. Other simultaneous EWS of both indicators are found for different scale ranges prior to DO-1, 2 and 6 in NGRIP and NEEM (see Supplementary Fig. S8 and S9).

   The number of significant increases of $\hat{w}^2$ and $\hat{H}$ for frequency bands relevant for the individual records are shown in Fig. A5. It reveals that the numbers of significant increases in $\hat{w}^2$ are statistically significant at 95% (considering the binomial distribution) for NGRIP and NEEM with temporal resolutions $\leq 10$ years for most scale bands $(s_1, s_2)$ with $s_1 \leq 30$ years. These

numbers are not significant for the records with coarser temporal sampling and any scale bands considered. Nevertheless, there are such bands for NGRIP in all considered resolutions and NEEM where simultaneous increases in $\hat{w}^2$ and $\hat{H}$ occur. Out of all the cases considered, only one $\delta^{18}$O record displays a significant number of significant increases in $\hat{H}$ for one scale band (GRIP, 60–100 years). Further synopses of the wavelet-based indicators across the various records and scale ranges are depicted in the Supplementary Sect. S1.3.



**Figure 9.** Early warning signals in the wavelet-based indicators confined to the 20–60-year periodicity band of various Greenland $\delta^{18}$O records. (a) Time series of the scale-averaged wavelet coefficient (black) during GS. (b) Same as **(a)** but for the local Hurst exponent. (c-d) Same as (a-b) but for the 10-year NGRIP record. (e-f) Same as (a-b) but for the 10-year NEEM record. (g-h) Same as (a-b) but for the 20-year NGRIP record. (i-j) Same as (a-b) but for the 20-year GRIP record. (k-l) Same as (a-b) but for the 20-year GISP2 record. Line colours and shadings are applied in the same way as in Fig. 4.





### 3.7 Summary of results

Throughout our analysis we found varying numbers of significant EWS across different indicators and $\delta^{18}$O records. These appear to be statistically significant primarily for NGRIP and NEEM with a temporal resolution $\leq 20$ years and the indicators of high-frequency variability $\sigma^2$ and $\hat{w}^2$. Considering the wavelet-based estimators, it appears that the choice of wavelet basis plays a critical role in whether a significant amount of EWS is observed.

An overview of the number of significant EWS, as well as their statistical significance, across the different ice core records and a selection of indicators is shown in Fig. 10.





**Figure 10.** Numbers of significant EWS and their statistical significance at 90 and 95% across various $\delta^{18}O$ records from Greenland ice cores and a selection of indicators. Ice core records are denoted by their location and temporal resolution. The wavelet-based indicators $\hat{w}^2_{s_1,s_2}$ and $\hat{H}_{s_1,s_2}$ are specified by the choice of wavelet basis, and the considered scale ranges between $s_1$ and $s_2$ years.

### 3.7.1 NGRIP

For the NGRIP record interpolated to 5-year time steps, we find fewer significant EWS compared to Boers (2018), when significance testing is altered. Further changes in EWS indicator calculation and data preprocessing were found to have a minor influence. Nonetheless, our results also indicate a lower number of spurious early warnings in all the estimators considered. We observed a strong agreement between the binomial and numerically constructed null-distributions of false positives. Hence, we





argue that our surrogate model, used to determine whether an EWS is significant, better represents the null hypothesis of DO events occurring due to random fluctuations.

The numbers of significant increases for this record are statistically significant at 95% confidence for the variability estimators $\sigma^2$ and $\hat{w}^2$ using the Paul wavelet function, as well as simultaneous occurrences of CSD- and wavelet-based EWS, respectively. However, we note that these simultaneous EWS do not occur before the same transitions for the CSD- and wavelet-based indicators (DO-12 and 1, respectively) and not all variability indicator increases are preceding the same DO events. The choice of wavelet basis function for the calculation of the wavelet-based indicators was found to be critical for the detection of significant EWS, where increases prior to DO-1, 6 and 12 appeared to be less sensitive.

Applying specialised approaches for irregularly sampled time series to the raw NGRIP record yields similar results as for the regularly-sampled one. However, we could observe a tendency towards more significant EWS. The number of significant increases is statistically significant at 95% confidence for all indicators considered, except $\hat{H}$.

### 3.7.2 Comparison of ice core records

Most transitions do not show consistent EWS across various $\delta^{18}$O records with regular time steps from different Greenland ice cores, with the notable exception of DO-1, and to a lesser degree DO-6 and 12, which agree in EWS of the variability indicators in the high-resolution records from NGRIP and NEEM. These are also the only records displaying a significant number of significant EWS in the variability estimators. We find fewer EWS and less agreement for GRIP and GISP2. For the estimators of correlation times, we only find a statistically significant number of EWS in $\alpha_1$ in the GISP2 record, that otherwise doesn't display a significant number of significant indicator increases. Only few significant increases in $\hat{H}$ are seen, and the observed numbers are not statistically significant at 95% confidence for any of the records.

## 4 Discussion and conclusions

### 4.1 Implications of results

In comparison to the results by Boers (2018), our analysis reveals fewer significant EWS for individual DO events in the high-frequency variability of the high resolution NGRIP record. Only few events, notably DO-1,6 and 12, are preceded by consistent significant increases across the different variability indicators and the various $\delta^{18}$O records studied here. While multiple previous studies also found significant EWS for DO-1 (Rypdal, 2016; Boers, 2018; Myrvoll-Nilsen et al., 2024) and DO-12 (Rypdal, 2016; Boers, 2018) only the results from Boers (2018) indicate a destabilisation prior to DO-6.

We found a statistically significant amount of significant early warnings in the CSD- and wavelet-based indicators or high-frequency variability, especially for $\delta^{18}$O records with higher temporal resolution. However, due to lack of consistent accompanying EWS in correlation times, we find only weak evidence for a destabilising climate state prior to these or any of the DO transitions, which would be expected if they were bifurcation-induced. One reason for the fewer observed and less consistent significant EWS in $\sigma^2$ and $\hat{w}^2$ for NGRIP, GRIP and GISP2 sampled every 20 years might be that their resolution is too coarse





to study imprints of processes on (sub-)centennial time scales. We further note that the differences between the NGRIP record in different resolutions may be caused by sampling effects and/or a result of spurious EWS.

We do not find clear support for the hypothesis that any of the analysed transitions are caused by a bifurcation in a dynamical subsystem operating at decadal time scales, as proposed by Rypdal (2016) and previously confirmed by Boers (2018). It is important to note that our findings cannot be used to reject such a hypothesis either, and that the observed precursor signals do not directly yield an indication on which mechanisms might be most relevant for DO events.

The indicators used in this study are based on relatively simple low-dimensional dynamical systems characterized by specific
bifurcation and noise structures (Scheffer et al., 2009; Ditlevsen and Johnsen, 2010; Kuehn, 2011). However, they may not produce equivalent results when applied to observational data from more complex systems, such as the Earth's climate, which features more intricate bifurcation structures, varied noise processes, and many interacting time scales. This suggests the need for a more cautious approach, one that is specifically tailored to the unique properties of the underlying system – assuming these properties are well-understood. Consequently, gaining a deeper insight into the processes driving DO cycles becomes
essential. Recent advancements in EWS methods have expanded to address various noise processes (Kuehn et al., 2022; Morr and Boers, 2024) and introduced new methodologies (Clark et al., 2002). However, there remains a significant need for further research into the applicability of EWS.

Despite the simplicity of the EWS used in our analysis, we faced numerous decisions regarding parameters, significance tests, and computational details. These choices can substantially influence results, as evidenced by our comparisons with the
findings of Boers (2018) (Figs. 4 and 7) and our adaptations for analysing irregularly sampled time series (Figs. 5 and 8). This highlights the sensitivity of the results to these methodological choices and underscores the need for careful consideration and a comprehensive understanding of when and how these methods might be beneficial.

Furthermore, it is important to recognise the limitations of EWS, such as the potential for false positives and their inconsistent ability to predict transitions in complex systems, as demonstrated by our analysis of "obvious" transition scenarios where EWS
did not provide reliable foresight. This calls into question the reliability of EWS in predicting future system behaviors and emphasises the need to approach their use with caution. The situation here is further complicated by applying such indicators to the temperature proxy $\delta^{18}O$ from ice core records, which in itself is subject to a multitude of influences, some of which will be discussed below.

Due to the observed inconsistencies in high-frequency fluctuations across the different records, we note that some of the
observed "early warnings" may not stem from a common climate background, but are likely caused by other factors specific to the ice cores' locations, while others might be masked for the same reasons.

## 4.2   Differences between the ice core records

While the $\delta^{18}O$ records from the four different ice core sites all show the same synchronous behaviour during GS/GI transitions (Guillevic et al., 2013; Seierstad et al., 2014)(see Fig. 2), they differ in some aspects besides their resolution.

GRIP and GISP2 are located approx. 28 km km apart from each other on the summit of the Greenland ice sheet (Guillevic et al., 2013), whereas NGRIP was drilled on the ice divide, approx. 325 km north-west of GRIP (North Greenland Ice Core



Project members et al., 2004). NEEM lies ca. 350 km further north-west along this divide (Erhardt et al., 2022). The locations
of these ice core sites are depicted in Fig. 1.

It has been shown before that $\delta^{18}$O values are systematically between 1 and 3 ‰ lower in NGRIP compared to GRIP
and GISP2 throughout the last glacial period (North Greenland Ice Core Project members et al., 2004; Guillevic et al., 2013;
Seierstad et al., 2014). We also note that these values are comparable between NGRIP and NEEM (Guillevic et al., 2013), and
the two summit cores (Seierstad et al., 2014), respectively. For DO-8 and 10, Guillevic et al. (2013) found that the difference
in the water isotopic ratio $\delta^{18}$O between GS and GI decreases from North Western Greenland to its summit. Given their
geographical proximity, the discrepancies between the signals are remarkable and indicate important regional variations (North
Greenland Ice Core Project members et al., 2004).

Oxygen isotope ratios $\delta^{18}$O are often used as temperature proxies of the past (Dansgaard, 1964), but they are also influenced
by complex effects from the mixing of air masses (Charles et al., 1994). Important factors are the distance and temperature
gradient between the ice core and its source region of precipitation (Jouzel et al., 2000; Steen-Larsen et al., 2013), as well as
seasonality biases of the received precipitation (Krinner et al., 1997; Werner et al., 2000; Langen and Vinther, 2009; Seierstad
et al., 2014). Indeed, for DO-8, 9 and 10, the temporal sensitivity of $\delta^{18}$O to temperature was found to vary from 0.34 to
0.68 ‰ °C$^{-1}$, where it decreases with site elevation, i.e from NEEM to the summit sites (Guillevic et al., 2013).

Possible reasons for the spatial inhomogenities between the records include changes in moisture origin and transport paths,
precipitation seasonality, meso-scale atmospheric dynamics and local processes (Guillevic et al., 2013; Seierstad et al., 2014;
Capron et al., 2021; Steen-Larsen et al., 2013). Differences between the records on shorter (sub-millennial) time scales are
thought to have been driven by rapid sea ice and/or sea surface temperature changes in the North Atlantic, which were found to
have a stronger influence on the $\delta^{18}$O variability in North-West Greenland than on the summit (Guillevic et al., 2013; Seierstad
et al., 2014). Multiple previous studies suggest that DO events in Greenland were triggered by a rapid sea ice retreat in the
North Atlantic (Broecker et al., 1985; Ganopolski and Rahmstorf, 2001; Gildor and Tziperman, 2003; Li et al., 2010; Dokken
et al., 2013; Petersen et al., 2013; Zhang et al., 2014; Hoff et al., 2016; Vettoretti and Peltier, 2016; Boers et al., 2018; Vettoretti
and Peltier, 2018; Li and Born, 2019; Riechers et al., 2023a). The influences of those changes on $\delta^{18}$O values may therefore
be more pronounced in the NGRIP and NEEM records, potentially contributing to the more frequent and consistent presence
of EWS in these records.

Another factor that might play into the similarity of results between the two records from the Greenland divide could be that
the NEEM ice core is located downstream of NGRIP (Dahl-Jensen et al., 2013; Montagnat et al., 2014). It has been shown
before that the current NGRIP site was located at a higher altitude and further upstream, closer to NGRIP than it is today
(Dahl-Jensen et al., 2013), whereas past NGRIP deposition sites were situated fairly close to its present-day location (Nixdorf
and Göktas, 2001) due to a constant horizontal velocity along the ridge around NGRIP (Dahl-Jensen et al., 2002).

Another inconsistency across the sites are snow accumulation rates. The two summit sites are believed to have similar
accumulation histories, with higher rates than at NGRIP and NEEM (Guillevic et al., 2013; Seierstad et al., 2014). A previous
study (Münch et al., 2016) on Antarctic ice cores indicates that in $\delta^{18}$O records from locations with low snow accumulation, the
highest frequencies may predominantly be influenced by disturbances occurring after deposition. While the sites studied there



generally display substantially lower accumulation than the Greenland sites, it is important to note that Greenland accumulation rates decrease to comparable low values during GS (Guillevic et al., 2013; Seierstad et al., 2014; Münch et al., 2016). Hence, we cannot rule out that the observed EWS are dominated by such intrinsic noise, even though their simultaneous occurrence and statistically significant numbers in the high-resolution NGRIP and NEEM records seem to indicate otherwise.

The reduced number of significant EWS for DO-1 in GRIP and GISP2, compared to NGRIP and NEEM might be explained by important uncertainties in the time scale transfer from NGRIP during long stadials, such as GS-2.1 preceding DO-1 (Seierstad et al., 2014). Regardless, even larger uncertainties were estimated for NEEM during the same period (Rasmussen et al., 2013).

Possible reasons for the differences in results for the GISP2 record might be related to the missing values in the $\delta^{18}$O time series (see Sect. 2.1 for details). Moreover, parts of this record had to be corrected for alterations of $\delta^{18}$O by the way some of the ice core samples have been stored (Stuiver et al., 1995). Nonetheless, these corrections were later found to have a minor influence on parts the record (Stuiver and Grootes, 2000). Those inconsistencies might further be related to the fact that $\delta^{18}$O from NGRIP, NEEM, and GRIP has been measured at the University of Copenhagen (North Greenland Ice Core Project members et al., 2004; Gkinis et al., 2014; Johnsen et al., 1997; Gkinis et al., 2021), whereas the GISP2 has been analysed at the University of Washington (Stuiver and Grootes, 2000).

The aim of this discussion is not to give a comprehensive overview of possible drivers of differences in $\delta^{18}$O records from Greenland ice cores. Instead, it serves to illustrate that there is a diverse range of factors, other than a common high-frequency climate signal, that could have major influences on the results presented here.

*Code and data availability.* The raw NGRIP data, as well as data from NGRIP, GRIP, and GISP2 resampled to 20 year resolution are freely available at http://www.iceandclimate.nbi.ku.dk/data/. The raw NEEM data can be found at https://doi.org/10.1594/PANGAEA.925552. All julia code used for the analysis of the $\delta^{18}$O records with regular temporal resolution, as well as python code for the analysis of the raw NGRIP record is available upon request via email (c.hummel@uit.no).



# Appendix A: Additional figures

**A1    Resampling of irregularly sampled data**

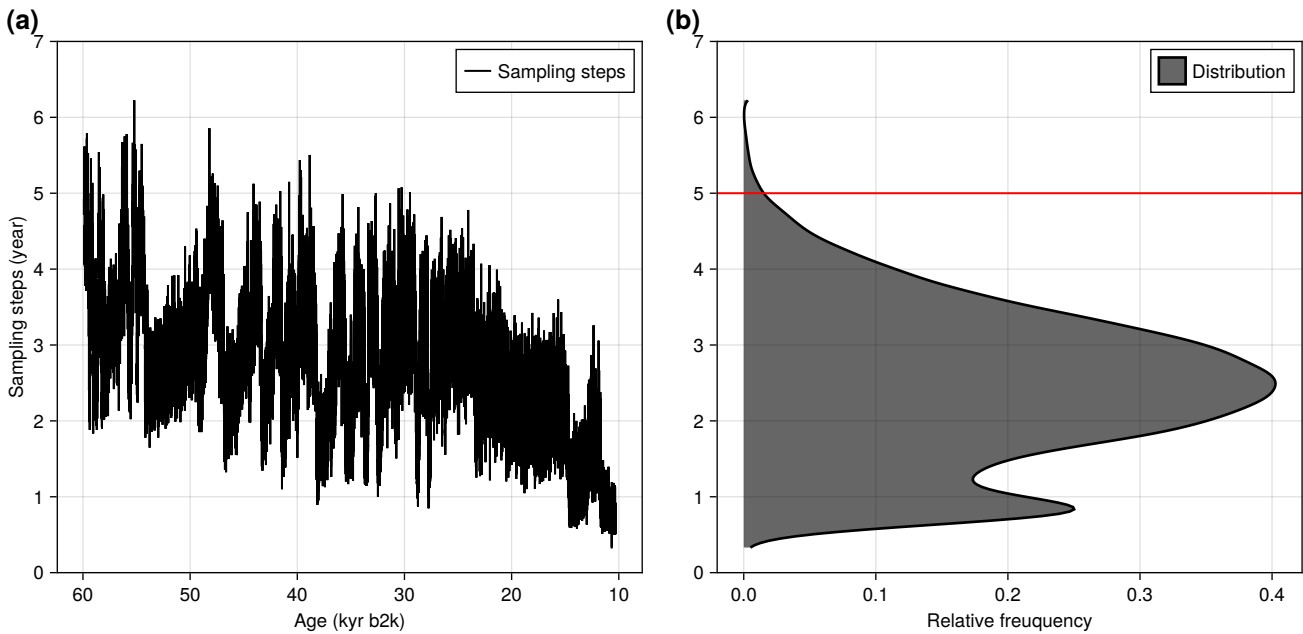

**Figure A1.** Temporal sampling steps of the raw NGRIP $\delta^{18}$O record on the GICC05 time scale as a function of time (a) and their distribution (b). The horizontal red line marks the temporal resolution of 5 years.



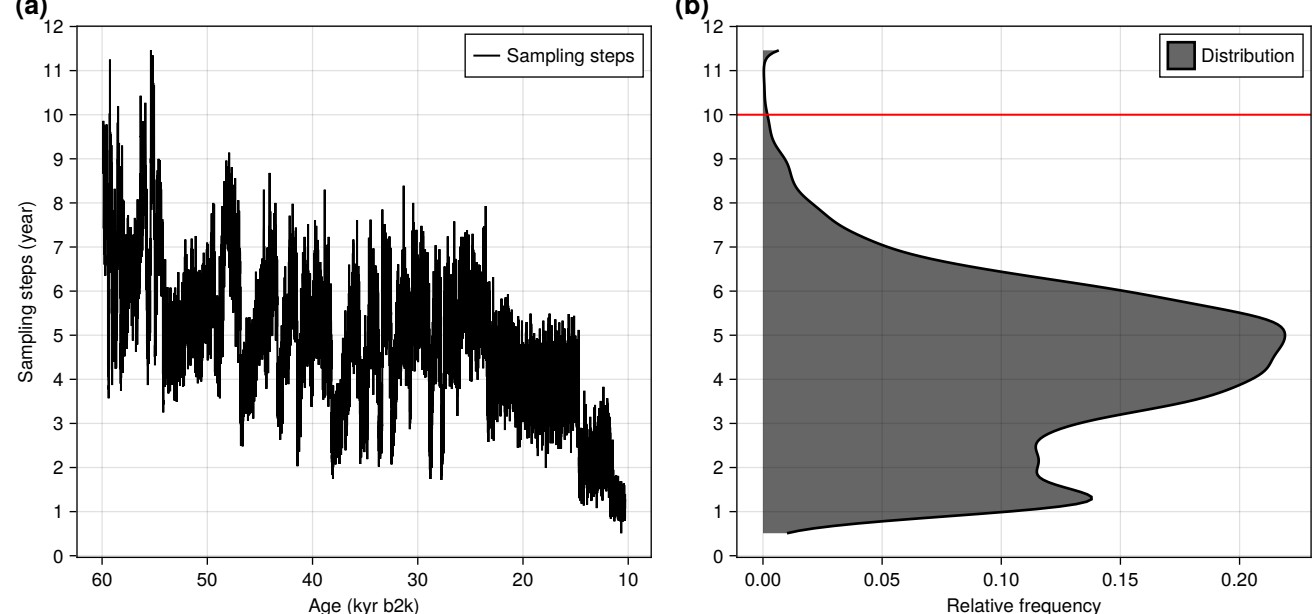

**Figure A2.** Temporal sampling steps of the raw NEEM $\delta^{18}$O record on the GICC05 time scale as a function of time (a) and their distribution (b). The horizontal red line marks the temporal resolution of 10 years.





## A2 Significance testing

**Figure A3.** Null-model distributions for the number of significant EWS in $\hat{w}^2$ (a), $\hat{H}$ (b), and both wavelet-based indicators simultaneously (c) for the NGRIP $\delta^{18}$O record with 5-year resolution.




## A3 Wavelet-based EWS

**Figure A4.** Linear trends of wavelet-based early warning indicators in a selection of scale bands $(s_1, s_2)$ for individual transitions of the NGRIP record in 5- (a-c), 10- (d-f) and 20-year resolution (j-l), NEEM (g-i), GRIP (m-o), and GISP2 (p-r). The direction of trends of the scale-averaged wavelet coefficient are shown in the left (a,d,g,j,m,p), those of the local Hurst exponent in the middle column (b,e,h,k,n,q). The right column (c,f,i,l,o,r) shows an increasing trend if both indicators increase and a decreasing trend otherwise. Significant indicator increases are displayed in dark red.







**Figure A5.** Numbers of significant wavelet-based EWS in different scale bands between $s_1$ and $s_2$ years of the NGRIP record in 5- (a-c), 10- (d-f) and 20-year resolution (j-l), NEEM (g-i), GRIP (m-o), and GISP2 (p-r). EWS of the scale-averaged wavelet coefficient are shown in the left (a,d,g,j,m,p), those of the local Hurst exponent in the middle (b,e,h,k,n,q) and of both simultaneously in the right column (c,f,i,l,o,r).



*Author contributions.* CH, MR, and NB conceived the study. CH performed the analysis. All authors discussed and interpreted results. CH prepared the manuscript with contributions from all authors.

*Competing interests.* The authors declare that they have no conflict of interest.

*Acknowledgements.* This project has received funding from the European Union's Horizon 2020 research and innovation programme under the Marie Skłodowska-Curie grant agreement No.956170 (CriticalEarth). It has further been supported by the Research Council of Norway (project number 314570) and the UiT Aurora Centre Program. This is ClimTip contribution #X; the ClimTip project has received funding from the European Union's Horizon Europe research and innovation programme under grant agreement No. 101137601. NB acknowledges

further funding from the Volkswagen Foundation.



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
