# Peer review of "Inconclusive Early Warning Signals for Dansgaard-Oeschger events across Greenland ice cores"

_EGUsphere, 2024_

## Author Comment (AC1)

General Comments:

This is an extremely thorough and comprehensive analysis of whether Early Warning Signals (EWS) are present before Dansgaard-Oeschger (DO) events in Greenland ice cores. Importantly, this study rigorously addresses the methodological limitations and discrepancies that have led to conflicting results in previous studies on this topic. The statistical approach used to detect EWS is justified by the close match between the analytical and numerical distributions for the number of false positives. Although this detection of significant EWS preceding individual transitions is carried out extremely well, I believe there is an issue with the analysis of whether the number of observed EWS is in turn significant itself. This could impact the findings and therefore needs to be carefully addressed. Overall, I recommend that this manuscript is published subject to revisions.

We thank John Slattery for the very helpful and thorough review. The comments will be addressed point-by-point below.

Specific Comments:

The only major issue in this otherwise excellent manuscript concerns the analysis of whether the number of observed EWS is significant. As the authors correctly state on line 218: "*For x ~ B(17, 0.05), it is P(x ≤ 2) ≈ 0.9497 < 0.95 and P(x ≤ 3) ≈ 0.9912 > 0.95.*" However, the authors then mistakenly infer from this that "*at a confidence level of 95%, we expect at most two events to show spurious significant early warning, and observing three significant EWS is statistically significant.*" In fact, the number of EWS required for statistical significance at the 95% level is N, where N is the smallest integer such that, for x ~ B(17, 0.05), P(x < N) > 0.95. The crucial difference is that the probability of x being less than **but not equal to** N must be more than 95%, not less than or equal to as the authors imply. The significance threshold at the 95% confidence level using this analytical distribution is therefore four significant EWS observed, not three.

One can consider this in an equivalent way that may be clearer by thinking instead about the p-value as compared to the significance level (i.e. 1 - confidence level). A result is significant at the 5% significance level if, under the null hypothesis, the probability p of observing a result at least this extreme is less than 5% (i.e. p < 0.05). In our case, the number of observed EWS required for significance is N, where N is the smallest integer such that, for x ~ B(17, 0.05), P(x ≥ N) < 0.05. P(x ≥ 3) = 1 - 0.9497 = 0.0503 > 0.05, and so observing three EWS is not quite statistically significant at the 5% / 95% level, whilst observing four is.

To see clearly that the authors' approach is mistaken, consider Figure 3(a&b). Both the analytical and numerical distributions show that there is a 16% probability of 2 out of the 17 transitions showing false positive EWS. Despite this, the authors indicate in 3b that observing two EWS is significant at the 95% level using the numerical distribution. Elsewhere, including Figure 10, they also indicate that observing two EWS is significant at the 90% level for both distributions. Observing two EWS cannot be significant at either

confidence level or with either distribution, though, because this happens by chance 16% of the time! For another example, consider the distribution for simultaneous EWS in Figure 3c. Using the authors' logic, $P(X \geq 0) > 0.95$ and so 0 transitions with simultaneous EWS in both indicators would be a significant positive result, which clearly cannot be the case. I hope that these examples demonstrates that my comment here is not merely a statistical foible or a petty criticism, but that it has a real impact on the findings of this study.

We thank the referee for pointing this out and agree that the approach we used to calculate significance thresholds for the number of significant EWS was indeed wrong. The statement in lines 218-220 will be corrected accordingly and the significance thresholds will be corrected in the revised manuscript and all relevant figures (i.e. Figures 3, 10, A3, S14, S16, and S19).

The comparison of the analytical and numerical distributions is a fantastic way to show that the significance test for EWS preceding individual transitions works as intended, and I applaud the authors for including this. However, having done so, I think it would be better to then consider only the significance threshold for the number of observed EWS derived from the analytical distribution. This would simplify the analysis by making the threshold the same for all records and indicators. Currently there is sometimes (e.g. Figure 3b) a discrepancy between the thresholds for the two distributions, even though they match very well, just because $P(x < 3)$ is so incredibly close to 0.95.

We agree that the analysis would be simplified by only using the analytical significance threshold. Hence, we will remove references to the numerical distribution when comparing the different records and summarising the results.
Nevertheless, we note that numerical distributions have only been calculated for the NGRIP record with 5-year resolution, and argue that the different significant thresholds stemming from the numerical and analytical distributions are worth mentioning in this case. Thus, they will only be mentioned briefly in the main text of Sections 2.4 (Methods - Expected number of spurious significant EWS) and 3.7 (Results) in the revised version of the manuscript.

Line by line comments:

Line 22 and elsewhere: This study describes $\delta_{18}O$ in Greenland ice cores as a local temperature proxy, following the traditional interpretation. However, recent isotope-enabled modelling (Buizert et al. 2024, https://doi.org/10.1073/pnas.2402637121) suggests that winter sea ice variation may instead be the dominant control on $\delta_{18}O$ during DO events. I suggest that this new interpretation should be briefly discussed, either in the introduction or in Section 4.2.

We greatly appreciate this input and will include a short discussion on this in Section 4.

Figure 3: I think it would also be better to place the significance threshold lines between integers, as it is currently unclear whether observing a number of EWS equal to the significance threshold is significant or not. Indeed, Figure 3c seems to suggest that the

significance threshold is 0, if interpreted in the same way as a & b, which cannot be the case. This should of course also account for the corrected significance thresholds based on my main comment, and the same also applies to Figures A3 and S19.

*We agree and will place the (corrected) significance threshold lines between integers in Figures 3, A3, S16 and S19.*

Lines 253-254: "*Furthermore, we don't restrict the search for wavelet-based EWS to the GS until 200 years prior to events to include potential influences of the transitions themselves.*" This sentence is unclear to me.

*This sentence will be changed to "Furthermore, we don't restrict the search for wavelet-based EWS to the GS until 200 years prior to events, as in Boers (2018), to exclude potential influences of the transitions themselves. Instead …"*

Lines 287-288: "*Though, observing two significant EWS in $\alpha_1$ is only significant with respect to the analytical, but not the numerical null-distribution.*" Based on Figure 3b this appears to be the wrong way round, as the numerical threshold is two EWS whilst the analytical threshold is three. Either way, as mentioned above, I think it would simplify the analysis to consider only the analytical null distribution.

*We agree that this is the wrong way round. Following the comment above, this statement will be corrected, including only the corrected analytical threshold.*

Figure 10: It is difficult to distinguish between zero and undefined using this colour scheme. The circles indicating significance should also be corrected as discussed above.

*We will change the colour scheme and adjust the significance thresholds.*

Lines 480-481: "*Recent advancements in EWS methods have … introduced new methodologies (Clark et al., 2002).*" It seems odd to call a study from 23 years ago a recent advancement. Perhaps a different reference was intended here, otherwise this sentence should be reworded slightly.

*We agree. A different reference was intended here indeed. It will be replaced with (Clarke et al., 2023, https://doi.org/10.1088/1748-9326/acbc8d).*

Lines 529-530: "*It has been shown before that the current NGRIP site was located at a higher altitude and further upstream, closer to NGRIP than it is today.*" This sentence is unclear. I think that the authors perhaps intended to write NEEM here instead of NGRIP.

*We indeed intended to write NEEM instead of NGRIP and will correct this sentence to "It has been shown before that the current NEEM site was located at a higher altitude and further upstream, closer to NGRIP than it is today [...]."*

Fig S3c in Supplementary Information: The line for the 95% confidence interval is either hidden or missing.

We will recreate this figure, including the hidden line for the 95% confidence interval.

Technical Comments:

We appreciate all the technical comments below and will correct these mistakes before resubmission.

Line 202: *"allows to handle data"* is missing a word. This should perhaps read "allows us to handle data".

Line 287: *"…and autocorrelation for DO-12 Though, observing two significant EWS…"*. I think there ought to be a full stop between *"DO-12"* and *"Though"*.

Line 314: *"resolutions.Another"* is missing a space after the full stop

Figure 7 caption: *"(e-f) Same as (c-d) but with modified estimator calculation.(g-h) Same as (e-f) but with modified data preprocessing.Line colours and shadings are applied in the same way as in Fig. 4."* Spaces are missing after both full stops.

Line 459: *"notably DO-1,6"* is missing a space after the comma.

Lines 519-520: *"(Guillevic et al., 2013; Seierstad et al., 2014; Capron et al., 2021; Steen-Larsen et al., 2013)"* These references are not in chronological order.

Line 548: *"on parts the record"* is missing a word.

I hope that these comments are helpful, and I look forward to reading the authors' response.

We are grateful for these detailed and valuable comments and thank John Slattery for his input and feedback.

---

## Author Response (AR1)

**Author's response to reviews of "Inconclusive Early Warning signals for Dansgaard-Oeschger events across Greenland ice cores"**

**Review 1:**

This paper presents a thorough analysis of Early Warning Signals (EWS) prior to the abrupt Dansgaard-Oeschger events observed in Greenland ice-core records. All the available deep records, GRIP, GISP2, NGRIP and NEEM are used for the analysis. EWS are changes in statistical properties of a time series indicating a bifurcation-induced transition (b-tipping), they will not appear prior to a noise-induced transition (n-tipping). The aim is thus to identify for each of 17 DO-events in the well-dated past 60kyr records which would be due to b-tipping and which would be due to n-tipping assuming a classical bistable dynamics. As the detailed dynamics of the transitions are largely unknown, the simplest assumption (Occam's razor type of argument) is that of a saddle-node bifurcation in a system subject to noise. In such a system variance will, from the fluctuation-dissipation theorem, increase when approaching the bifurcation point, likewise will the autocorrelation increase. This is the phenomenon of critical slow down. For any other suggested scenarios for the transitions, different EWS could potentially be detected. Since the transitions documented in the paleoclimatic records have already happened, detected EWSs obviously play the roles of hindcasts rather than forecasts, thus the purpose of detecting EWSs is rather dynamical system identification.

A fair statistical significance test is constructed by booth-strapping through generation of so-called Truncated Fourier Transform Surrogates (TFTS), which is just surrogate timeseries constructed by randomly choosing phases (not shuffling) of the Fourier-coefficients while keeping the amplitudes of the original signal. "Truncated" refers to not changing phases of the long wavelength coefficients to preserve trends in the timeseries. Since the variance and the autocorrelation in a time series only depends on the amplitudes of the Fourier coefficients, the TFTS will have the same variance and autocorrelation as the original time series over the full glacial state (GS) period analyzed. The EWS indicators are now calculated within 200y running windows for each of the GS periods prior to the DO-transitions and the slope of the linear fit of this indicator time series is calculated and a significant slope (at the 95% confidence level) is identified from the distribution of slopes in the TFTS time series. From this analysis it is established that only a few DO-events are preceded by EWS, in agreement with the expectation that about one of the 17 DO events should be significant at the 95% confidence level, motivating the title of the paper.

The findings confirm our earlier findings (Ditlevsen and Johnsen, 2010), so in some sense this is a reporting of negative results. However, I find that the paper presents useful methods for this kind of analysis, thus I recommend publication. I do though recommend a revision for clarifications and better readability:

Answer: We thank Peter Ditlevsen for this helpful and thoughtful review. All comments are addressed in our point-by-point responses below.

1. The GS vary in duration, a typical GS lasts perhaps 2ky, which means that there are ten independent 200yr window measurements. Thus, the linear trend is made for only 10 points or maybe even less. Furthermore, for the 20yr resolution records, there are only ten points within a 200yr window, from which the EWS are calculated. A discussion of the uncertainty and the quality of the estimates is lacking.

Answer: We agree that the individual stadials provide rather short time series to conduct EWS analyses on. While there are indeed only few independent, non-overlapping window measurements, we note that we use sliding windows to compute the CSD EWS indicators. So, for a typical stadial of 2ky, the linear trend of EWS indicators is calculated from 360 (180, 90) data points for records in 5-(10-, 20-) year resolution.

By comparing records from different ice core locations with varying temporal resolutions, as well as various methodological choices, our analysis shows that the EWS indicators used here are indeed sensitive to these factors. As such, this gives an insight into the uncertainties of EWS indicators and provides a more comprehensive view than previous studies on EWS preceding DO events. We think that the uncertainty and the quality of the EWS estimates is carried out in detail regarding the results of the significance tests, both with respect to individual trends, and with respect to the number of significant trends. Also note that the significance test for the trends is based on phase surrogates of the underlying stadial sections and thus incorporates the length of these time series segments. We will address this in the revised manuscript.

- Sentence in I. 245-250 added: "By taking surrogates for each individual GS with the same length as the δ18O record during that interval, we derive null-distributions for each stadial and record individually. Hence, our statistical significance test is adapted to the varying length of GS."
- Reason added for taking surrogates during GS only in Table 2: "Account for GS length"
- The discussion on the sensitivity of EWS indicators has further been expanded in I: 601-606: "[...] our results merely show that the presence or absence of significant EWS prior to DO events depends on various factors, such as the choice of the ice core, the resolution of the ice core record, specific data processing, choice of indicator, computational details and significance testing, giving insight into the uncertainties of EWS indicators. We thus highlight that EWS for DO events in particular, and applied to observational data in general, can be sensitive to uncertainties in the underlying time series, data preprocessing and methodological choices. This underscores the need for careful consideration and a comprehensive understanding of when and how these methods might be beneficial."

2. A consistency check between significant EWSs found for some, but not both EWS and some, but not all records (which are obviously false positives) and the number of false positives expected from the boot-strapping should be made.

Answer: We present the numbers of significant EWS for individual indicators and both occurring simultaneously for all ice core records. Further, we also constructed null-distributions for the number of false positives for individually and simultaneously increasing indicators, which indicate how many indicator increases are expected to occur by chance at different significance levels in a given record.

Nevertheless, because the records considered differ in many aspects, such as ice core location, processing and temporal resolution, we believe that observing EWS in some, but not all records does not necessarily imply a false positive. It could simply be that an underlying true EWS is masked in an individual core, with preprocessing steps affecting the different EWS indicators in different ways In order to ease the comparison, we will include a figure or table showing which DO event is preceded by which combination of EWS for each record and will comment more on this in the revised manuscript.

**Changes:**

- Sentence added in I. 570-573: "Since the δ18O records considered differ in many aspects, such as ice core location, processing and temporal resolution, observing significant EWS in some, but not all records does not necessarily imply a false positive. It could simply be that an underlying true EWS is masked in an individual record, with preprocessing steps affecting the different EWS indicators in different ways."
- Figures 9 now shows which DO event is preceded by which combination of EWS indicators
- 3. Figures 4-9 are difficult to read, unless they are only to be read like white-pink-red barcodes. Consider showing just one full width time series (for each EWS) and present the consistency between resolutions/methods/records in a figure similar to Figure S7 or S8, in order to get a better overview. There is a lot of information in the text, which makes reading difficult.

Answer: We agree that Figures 4-9 and the main text contain a lot of information. For the revised manuscript, we will also show our results in a more aggregated way to facilitate readability. Some of the current plots showing indicator time series will be moved to the supplementary material.

- Previous Figures 4-9 have been moved to the supplementary material (now Fig. S7-S-14) and replaced by Figures 6-8 presenting the results in a more aggregated way.
- Subsections in the results section (Sect. 3, starting in I. 304) have been merged to present all EWS indicators, both CSD- and wavelet-based, together for the NGRIP record with 5-year resolution (Sect. 3.1, I.305), irregular resolution (Sect. 3.2, I.382) and across ice core records (Sect. 3.3, I 411) and shortened accordingly.
- 4. Increase in the weights of specific wavelet coefficients and increases in local Hurst exponent, H, are suggested as EWS. However, it is not argued what the assumed underlying (complex) system exhibiting these EWSs are. There are references to earlier papers by the same authors (Rypdal, 2016, Boers, 2018), but these references do not provide such justifications. It is mentioned that the Hurst exponent is an estimate for the correlation, but if this is the only argument for calculating H, the authors should at least argue why it is more reliable to calculate H, than just analyze the autocorrelation (which is also done). It would substantially strengthen the paper if such arguments could be presented.

Answer: We agree that the presentation in our paper should be more self-contained, and will summarize the main reasons why the Hurst exponent is indeed a useful scale-aware extension of the AC1 coefficient in the revised version. We will explain why H is a scale-aware measure of the memory in the time series, which increases as a bifurcation is approached, motivating its use as an EWS. Moreover, since H has been used in previous papers, we believe it is important to include it here as well.

- Sentence added in I.195: "As an alternative approach to the commonly used CSD indicators V and \alpha\_1 described above, we also consider the scale-averaged wavelet coefficient \hat{w}^2 and the local Hurst exponent \hat{H}^{loc}, which have previously been applied as EWS for DO events (Rypdal, 2016; Boers, 2018)."
- Sentence and reference (Mei et al. 2023, https://doi.org/10.1175/JCLI-D-22-0263.1) added in I.211-213: "The local Hurst exponent \hat{H}^{loc} can be useful to describe how correlations decay in time, and is therefore expected to detect critical slowing down (Mei et al., 2023), given that it is estimated using a range of time scales that includes changes in the relevant processes."
- 5. A discussion of how a Hurst exponent can meaningfully be calculated from about one decade of time scales is lacking. What are the uncertainties?
  - Answer: We not 100% sure we understand the referee's concern here, but will clarify how exactly we compute the *local* Hurst exponent in our study; moreover,

uncertainties in the estimation of H are captured by our two significance tests, first based on phase surrogates to test individual trends, and second for the number of significant trends in the full time series.

**Changes:**

- We consistently name this indicator the *local Hurst* exponent (adapted e.g. in I. 132 and 294) and changed its notation from \hat{H} to \hat{H}^{loc} throughout the revised manuscript, including all figures.
- We did not change our description in how we calculate \hat{H}^{loc} since we believe it is detailed enough as it is (see Sect.2.2.2, I. 216-229)
- 6. In line 230 it is stated that for a linear stochastic process increase in variance and increase in autocorrelation are independent. This is not true: For the OU process x, we have Var(x) ~ -1/(log AC(1)).

Answer: We agree, of course. Nevertheless, the *estimates* of increases in variance and autocorrelation are independent under the null hypothesis that there are no parameter changes in the underlying system. This sentence will be reformulated for clarity accordingly.

**Changes:**

- The sentence has been changed (I. 266-268): "For a linear stochastic process not approaching a bifurcation, i.e. under the null hypothesis that there are no parameter changes in the underlying system, we would expect the estimates of increases in variability and correlation times to be independent"
- 7. The notation sigma^2 for Var(x) is unfortunate, since the underlying assumptions of the EWS is that the locally stationary process is the OU process: dx = -alpha x dt + sigma dB, where alpha\_1 = alpha dt (lag-1) AND Var(x)=sigma^2/(2 alpha). Thus sigma^2 represents the square of the intensity of the noise. I recommend using Var(x) for the variance.

Answer: We agree and the notation will be changed to Var(x) in the revised manuscript.

**Changes:**

• We changed the notation for the variance from \sigma^2 to V throughout the revised manuscript, including all figures.

I hope these comments are useful for the authors.

Answer: We highly appreciate these comments and thank Peter Ditlevsen for his input and feedback.

**Review 2:**

**General Comments:**

This is an extremely thorough and comprehensive analysis of whether Early Warning Signals (EWS) are present before Dansgaard-Oeschger (DO) events in Greenland ice cores. Importantly, this study rigorously addresses the methodological limitations and discrepancies that have led to conflicting results in previous studies on this topic. The statistical approach used to detect EWS is justified by the close match between the analytical and numerical distributions for the number of false positives. Although this detection of significant EWS preceding individual transitions is carried out extremely well, I believe there is an issue with the analysis of whether the number of observed EWS is in turn significant itself. This could impact the findings and therefore needs to be carefully addressed. Overall, I recommend that this manuscript is published subject to revisions.

Answer: We thank John Slattery for the very helpful and thorough review. The comments will be addressed point-by-point below.

**Specific Comments:**

The only major issue in this otherwise excellent manuscript concerns the analysis of whether the number of observed EWS is significant. As the authors correctly state on line 218: "For  $x \sim B(17, 0.05)$ , it is  $P(x \le 2) \approx 0.9497 < 0.95$  and  $P(x \le 3) \approx 0.9912 > 0.95$ ." However, the authors then mistakenly infer from this that "at a confidence level of 95%, we expect at most two events to show spurious significant early warning, and observing three significant EWS is statistically significant." In fact, the number of EWS required for statistical significance at the 95% level is N, where N is the smallest integer such that, for  $x \sim B(17, 0.05)$ , P(x < N) > 0.95. The crucial difference is that the probability of x being less than **but not equal to** N must be more than 95%, not less than or equal to as the authors imply. The significance threshold at the 95% confidence level using this analytical distribution is therefore four significant EWS observed, not three.

One can consider this in an equivalent way that may be clearer by thinking instead about the p-value as compared to the significance level (i.e. 1 - confidence level). A result is significant at the 5% significance level if, under the null hypothesis, the probability p of observing a result at least this extreme is less than 5% (i.e. p < 0.05). In our case, the number of observed EWS required for significance is N, where N is the smallest integer such that, for x  $\sim$  B(17, 0.05), P(x  $\geq$  N) < 0.05. P(x  $\geq$  3) = 1 - 0.9497 = 0.0503 > 0.05, and so observing three EWS is not quite statistically significant at the 5% / 95% level, whilst observing four is.

To see clearly that the authors' approach is mistaken, consider Figure 3(a&b). Both the analytical and numerical distributions show that there is a 16% probability of 2 out of the 17 transitions showing false positive EWS. Despite this, the authors indicate in 3b that

observing two EWS is significant at the 95% level using the numerical distribution. Elsewhere, including Figure 10, they also indicate that observing two EWS is significant at the 90% level for both distributions. Observing two EWS cannot be significant at either confidence level or with either distribution, though, because this happens by chance 16% of the time! For another example, consider the distribution for simultaneous EWS in Figure 3c. Using the authors' logic,  $P(X \ge 0) > 0.95$  and so 0 transitions with simultaneous EWS in both indicators would be a significant positive result, which clearly cannot be the case. I hope that these examples demonstrates that my comment here is not merely a statistical foible or a petty criticism, but that it has a real impact on the findings of this study.

Answer: We thank the referee for pointing this out and agree that the approach we used to calculate significance thresholds for the number of significant EWS was indeed wrong. The statement in lines 218-220 will be corrected accordingly and the significance thresholds will be corrected in the revised manuscript and all relevant figures (i.e. Figures 3, 10, A3, S14, S16, and S19).

**Changes:**

- The statement has been corrected (I.250): "For x ~ B(17, 0.05), it is  $P(x < 3) \approx 0.9497 < 0.95$  and  $P(x < 4) \approx 0.9912 > 0.95$ ."
- All statements in the relevant Sections (Sect. 2 Methods, Sect. 3. Results and Sect. 4. Discussion and conclusions), as well as the abstract have been corrected accordingly (e.g. 8-13, 251-252, 259-261, 313-314, 326-328, 391-393, 406, 426-429, 456-457, 462-463)
- All figures (Fig. 3, 6, 7, 8, 9, A3, S21, S22, S23, S24, S25, and S26) have been adapted to include the corrected thresholds

The comparison of the analytical and numerical distributions is a fantastic way to show that the significance test for EWS preceding individual transitions works as intended, and I applaud the authors for including this. However, having done so, I think it would be better to then consider only the significance threshold for the number of observed EWS derived from the analytical distribution. This would simplify the analysis by making the threshold the same for all records and indicators. Currently there is sometimes (e.g. Figure 3b) a discrepancy between the thresholds for the two distributions, even though they match very well, just because P(x

Lambda has been calculated as in Boers (2021) using 200-year moving windows and statistical significance of linear trends has been calculated as for the other indicators, using 10 000 TFTS surrogates.

We observe high correlation between the time series of lambda and the AR1 coefficient during most stadials. The Pearson correlation coefficients of lambda and the indicators used in our study during stadials preceding the abrupt transitions are shown below. We will add these results and a corresponding discussion in the revised manuscript.

**Changes:**

- The restoring rate lambda has been included in the introduction (I.68-69, 133-135)
- A description on why it is useful and how to calculate it was added to the methods section (Sect. 2.2.1, I.178-193)
- The results shown above for the NGRIP record with 5-year resolution, as well as a discussion of those, have been included into Sect. 3 Results:
  - Sect. 3.1 Early warning signals in the NGRIP record with 5-year resolution, from line 305, including Fig.4
  - $\circ$  Sect. 3.1.1 The restoring rate  $\lambda$ , from line 329, including Fig. 5
- 2. Regarding the method modifications wrt Boers (2018): Without being deep into the EWS methods, some of the modifications made in the present study may appear quite arbitrary. I suppose all changes were made to improve things or test a different equally plausible parameter choice. But I would ask the authors to explain the reasoning behind each step, why the tested method may be an improvement, or why it is important that a certain parameter or similar be tested.

Answer: We will add a column to Table 1 detailing the reasons why each of those modifications has been made.

**Changes:**

- Table 2 on p.14 now includes a column detailing the reasons why each of those modifications have been made
- 3. Regarding the respective modification steps:
- Did you test the methods only in sequence or also individually, e.g. step 3 without having performed step 1 and 2 first? Do you expect that the effects of all modifications add up linearly?

Answer: We also tested the modification steps individually and found that they did not add up linearly. Step 1 (changes to the significance testing) yielded the biggest decrease in the number of significant EWS, compared to steps 2 and 3, also when applied individually. The modifications are presented in sequence for easier readability.

**Changes:**

- We added a paragraph in Sect. 3.1., I.377-381: "The modifications shown here
  are applied in sequence. Nevertheless, we find that step 1 (changes to the
  significance testing) yields the biggest decrease in the number of significant
  EWS, compared to steps 2 (EWS calculation) and 3 (data preprocessing), also
  when applied individually."
- Why is step 3 the last thing you test? Should that not be the first (since you did not test different combinations of the modifications)? And given that step 3 does not seem to have any effect, could it not be omitted here for compactness (you might simply mention it in a side sentence)?

Answer: We start our modifications with step 1 (changes in significance testing) since we deem this to be the most important methodological change compared to Boers (2018). Steps 2 and 3 present comparably smaller changes. While step 3 does not seem to yield any effect, we would like to keep it included in order to provide a complete picture.

**Changes:**

- We added a sentence in Sect. 2.5.1, I.282-284: "We change how significance is tested as a first step in the sequence of different modifications since we deem this to be the most important methodological change compared to Boers (2018)."
- While we kept step 3 included in the revised manuscript, we note that the
  subsections in the results section (Sect. 3, starting in I. 304) have been merged to
  present all EWS indicators together for the NGRIP record with 5-year resolution
  (Sect. 3.1, I.305), irregular resolution (Sect. 3.2, I.382) and across ice core
  records (Sect. 3.3, I 411). For further compactness, we aggregated the
  information from previous Fig.4 and 7 into Fig. 6 in the revised manuscript.
- Step 1 has three items but they are all changed together? What has the biggest effect? The TFTS surrogates or the "entire time series" vs "GS-only"?

Answer: All modifications in step 1 are indeed changed together. These substeps combined yield a different method of testing significance than the one used in Boers (2018). We argue that only changing parts of this method would not give any further insights.

- We added a sentence in Sect. 2.5.1, I.281-282: "Since those modifications combined yield a different method of testing significance, they are not divided into sub-steps, as the changes in step 2 and 3 (see Table 2 and below), but are applied together."
- 4. Interpretation of the obtained numbers of EWS: The major part of the results seems to be a mere reporting of which method modification led to how many detected EWS.

Similar to comment 2, it would be good if the authors could provide some more interpretation of why the number of detected EWS changes in the different cases. And what the changes tell us about the respective transitions. Can we learn something about the nature of a transitions if it with one method an EWS is detected but with another not? Please provide some physical interpretation/context.

Answer: The main purpose of our study is to give a comprehensive account of EWS in Greenland ice core records, using all available ice cores and comparing different preprocessing steps and methods. This is important especially in view of previous papers, either reporting the absence or presence of significant EWS. Our results show that the EWS for the different DO events are not consistent across ice cores and methods, but also that more significant EWS arise than would be expected by chance. This could either be an unlikely but possible statistical outcome (given the null hypothesis of no EWS), or it could hint at biasing and masking effects of chemical processes happening in the ice core, and of the processing of the raw ice core data to obtain the proxy time series. Based on the available proxy data, it is, unfortunately, not possible to distinguish these two cases. In turn, the induced uncertainties and inconclusive results render a physical interpretation difficult. A clear presence of significant EWS across the different cores would give a strong argument for a bifurcation in the underlying dynamical mechanisms causing the DO events. But this is not the case as our results show, and we would therefore prefer not to go further into possible physical interpretations. Nevertheless, we think that the fact that based on a thorough and comprehensive statistical analysis, we can neither infer the absence of EWS prior to DO events, nor rule out their presence, is relevant and important to communicate, especially in view of diverting conclusions of previous studies.

Changes: none

5. Regarding the length of the Stadials: Does it matter that the intervals over which the ESW indicators are calculated are of different lengths? Trends will depend very much on the interval chosen, is that of relevance here? Also, the stadial between DO1 and DO2 contains the LGM, does it make sense to include it?

Answer: The sliding windows used to compute the EWS indicators have the same length for all stadials, but it is of course true that the stadials have different lengths. This is accounted for by computing the surrogates used to derive the distribution of the null hypothesis (no trend) individually for each stadial segment. In other words, the statistical significance test is adapted to the varying length of the stadials.

**Changes:**

• Sentence in I. 245-250 added: "By taking surrogates for each individual GS with the same length as the δ18O record during that interval, we derive null-distributions for each stadial and record individually. Hence, our statistical significance test is adapted to the varying length of GS."

- Reason added for taking surrogates during GS only in Table 2: "Account for GS length"
- 6. Regarding the broader picture: After your analysis, would you say there is a "best way" to estimate EWS? Do we now know more or less about EWS in general and ice cores in particular?

Answer: A best way to compute EWS can only be computed in experiments where it is known a priori that a given transition is induced by a bifurcation. This is not the case for the DO events and our results merely show that the presence or absence of significant EWS prior to DO events depends on the choice of the ice core, of the specific data processing, and of the EWS indicator. We thus know in more detail that EWS can be sensitive to uncertainties in the underlying time series and to data preprocessing steps. A key message of our study is hence a note of caution when applying EWS indicators in general. Specifically regarding Greenland ice cores, our results show that it cannot be safely concluded that the DO events are triggered by a bifurcation in the underlying dynamics, although it is important to note that the contrary can also not be concluded. We will expand the discussion along these lines.

**Changes:**

- The discussion has further been expanded in Sect. 4.2 Implications of results 598-606: "We further remark that our analysis does not aim to reveal a "best way" on how to calculate early warning signals. This can only be computed in experiments where it is known a priori that a given transition is induced by a bifurcation. This is not the case for the DO events and our results merely show that the presence or absence of significant EWS prior to DO events depends on various factors, such as the choice of the ice core, the resolution of the ice core record, specific data processing, choice of indicator, computational details and significance testing, giving insight into the uncertainties of EWS indicators. We thus highlight that EWS for DO events in particular, and applied to observational data in general, can be sensitive to uncertainties in the underlying time series, data preprocessing and methodological choices. This underscores the need for careful consideration and a comprehensive understanding of when and how these methods might be beneficial."
- 7. Regarding uncertainty: In Section 4.2, uncertainties in d18O are also discussed. Is it possible to take the proxy uncertainty into account? Or is it already being accounted for and I missed it? Does it matter that proxy noise is typically not white? Would EWS calculated for ensemble mean across ice cores be insightful and perhaps more robust?

Answer: Unfortunately, not all data used provides proxy uncertainties. Thus, a comprehensive analysis of those was not feasible.

Non-white proxy noise can potentially lead to false positives in the detection of EWS. Nevertheless, this is taken into account by our significance test, which can be seen from the close resemblance of the numerical and analytical null-distributions for the number of

**significant EWS.**

While an ensemble mean across ice cores could be interesting, we note that this would be a very small ensemble consisting of only 4 records from different ice cores. A larger ensemble including records in multiple temporal resolutions would be biased to those available in different and higher temporal resolutions (i.e. NGRIP and NEEM). Furthermore such an ensemble could not fully capture the differences between records. If we created an ensemble with records in the same resolution, this would be coarse, e.g. 20 years and the found tendency towards more EWS in high-resolution records (by us and Boers (2018)) could not be reflected. For an ensemble with records in their highest possible resolution, this tendency would also influence the ensemble mean. Thus, we believe that separate analyses of the different ice core records can provide more insights than an ensemble approach.

Furthermore, as mentioned in our reply to the previous comment, one of our main conclusions is a note of caution when applying EWS in general, including applications to potential future abrupt transitions. In those cases, ensembles of observations and proxies are typically not available.

Changes: none other than the added paragraph mentioned in our changes to comment 6 above.

**Minor comments:**

e.g. I.14 and I.122: be careful with words like "physical mechanism" and "climate background" - can the EWS really give insight into the physical mechanisms in terms of actual processes/feedbacks? Do you not rather mean the underlying tipping dynamics/bifurcation types?

Answer: We will carefully revise these sentences in the revised manuscript. We indeed mean the detection of signals of an underlying bifurcation, which we would call a mechanisms but we agree that a distinction to actual processes and feedbacks that might lead to such a bifurcation needs to be made.

**Changes:**

- "physical" has been removed describing causes and mechanisms not describing actual feedbacks and processes but rather tipping mechanisms, e.g. in I.2, 4, 13,
- "climate background" has been replaced by "climate signal", e.g. in I. 130, 563, 614

1.52-53 Consider moving sentence to beginning of paragraph starting 1.85

Answer: We agree and will move this sentence in the revised manuscript.

- The wording of this sentence has been slightly changed (I. 86-88): "Even though the background climate during the last glacial period and today are different, similar abrupt transitions as those during DO events may be triggered during current and future warming, where the transition may occur much faster than the change in forcing."
- It has been moved to the beginning of the proposed paragraph starting in I. 86

I.52/I.85 ff: The introduction opens the link between DO events and possible future AMOC transitions. Would this possible future transition not be more similar to the GI-GS transition, which is apparently very much understudied in terms of EWS? I very well understand that your focus is the GS-GI transition, but perhaps you could comment on this?

Answer: We will add a short discussion on this in the introduction.

**Changes:**

• We added a paragraph (I. 91-97): "A potential future weakening or shut-down of the AMOC would have severe impacts on the global climate and could lead to cooling over the Northern Hemisphere (Stouffer et al., 2006; Drijfhout, 2015; Jackson et al., 2015). Hence, future changes might be more comparable to past transitions from GS to GI, rather than DO events with changes from GI to GS, during the last glacial period. Past GS-GI transitions, as those shown in Fig. 2, occurred more gradually than the abrupt DO events and have consequently received less attention regarding possible EWS. Nevertheless, the presence of EWS for past abrupt transitions is the only empirical evidence that similar precursors may be found in observations before future tipping."

I.130-156: To me, it becomes not 100% clear which part of the resampling/interpolation of ice cores has been done for this study or already in previous studies. And regarding the NGRIP core(s), are the respective NGRIP cores independent cores or resampled/interpolated versions of the same irregularly sampled NGRIP core?

Answer: We considered resampled and interpolated versions of the same irregularly sampled NGRIP core. To make this clearer, a table will be added in the revised manuscript presenting the different cores, their resolution(s) and which previous studies considered them.

- I.221: "Previous EWS analyses for DO warming transitions have all been conducted on the δ18O record from the NGRIP ice core" instead of "records", and elsewhere
- I.126-128: "We conduct a systematic comparison of EWS during GS before DO events for a total of six δ18O time series from four ice core sites in three different temporal resolutions" instead of "six δ18O records"
- Table 1 has been added and gives an overview of the δ18O records considered, detailing the resampling method and giving references for these (either citations or to Sect. 2.1.2)

I.221-229 / I.188: What does it imply exactly, if the number of EWS is significant with respect to the analytical but not the numerical null-distribution? Is one a stronger/more meaningful constraint than the other?

Answer: The comparison of the two null-distributions primarily illustrates that our method of testing significance accurately represents the null-hypothesis. We don't deem either of the two to be more meaningful than the other. A sentence about this will be added. As also proposed by reviewer John Slattery, we will only consider the analytical null-distribution in Section 3 (Results) for simplicity and easier comparison between the different records, since numerical distributions have only been calculated for the NGRIP record with 5 year resolution.

**Changes:**

- Added a paragraph in I.262-265: "The comparison of the analytical and numerical null-distributions primarily illustrates that our method of testing significance (Sect. 2.3) accurately represents the null-hypothesis and we don't deem either of the two to be more meaningful than the other. In the following, we will primarily consider the binomial null-distribution for simplicity and easier comparison between the different records, since numerical distributions have only been calculated for the NGRIP record with 5 year resolution."
- The numerical distribution is now only mentioned in the methods section (Sect. 2.4, I. 253-265) and the summary of results for the NGRIP record (Sect. 3.4.1, I. 471-474)
- All other references to the numerical null-distributions have been removed.

1.269-273: The erroneous calculation was part of Boers (2018)?

Answer: Yes, we will rewrite this sentence to make this clearer.

**Changes:**

• Sentence in I.358-360: "The additional EWS in V stems from an erroneous calculation there, where the time series of the scale-averaged wavelet coefficient \hat{w}^2 was considered instead of the variance V."

1.277-278: Does this refer to the fact that the Boers (2018) curve is much smoother?

Answer: It does and it will be included in the revised manuscript.

**Changes:**

• Sentence in I. 366-368: "We note that the resulting indicator time series differ and appear less smooth because applying a 800-year low-pass filter, as done by Boers (2018) doesn't yield the same effect when applied to the GS rather than the entire time period (see Supplementary Fig. S7(a-d) and S9(a-d))."

I.504-506: would a systematic offset affect variance and auto-correlation?

Answer: Such an offset would not automatically affect variance and autocorrelation. Nevertheless, it is an important difference between the records. A sentence will be added for clarification.

**Changes:**

Sentence modified in I.508-510: "While these discrepancies between the signal
do not necessarily influence the EWS considered here, they are remarkable and
indicate important regional variations, given their geographical proximity (North
Greenland Ice Core Project members et al., 2004)."

I.506-508: Please always name the cores consistently. For readers not super familiar with the exact locations of the respective cores, it is difficult to immediately identify, which are e.g. the summit cores.

Answer: We agree and will adhere to a consistent naming of the ice cores in the revised version.

**Changes:**

- renamed the cores accordingly, e.g.:
  - o I. 505: "GRIP and GISP2" instead of "the two summit cores"
  - o I. 507-508, added " (i.e. from NEEM, over NGRIP towards GRIP and GISP2)"
  - I 516: "from NEEM over NGRIP to the summit sites GRIP and GISP2" instead of "from NEEM to the summit stes"
  - I.525-526: "variability in North-West Greenland (at NGRIP and NEEM) than on the summit (at GRIP and GISP2)
  - I. 532-533: "between the two records from the Greenland divide, NGRIP and NEEM. could be"
  - o I. 538: "GRIP and GISP2" instead of "the two summit sites"

I.528-532: I find this paragraph confusing, especially the second sentence. Please check the sentence and reformulate for clarity. The first sentence compares NEEM and NGRIP, the second sentence only mentions NGRIP, but four times. Is this correct?

Answer: We intended to write NEEM instead of NGRIP one of the four times in the second sentence and will correct it to "It has been shown before that the current NEEM site was located at a higher altitude and further upstream, closer to NGRIP than it is today (Dahl-Jensen et al., 2013), whereas past NGRIP deposition sites were situated fairly close to its present-day location [...]."

**Changes:**

• We changed this sentence (I.534-535): "It has been shown before that the current NEEM site was located at a higher altitude and further upstream, closer to NGRIP than it is today (Dahl-Jensen et al., 2013) [...]."

I.548-551: Would you not think, that a robust EWS should be detected, regardless of the lab that processed the core? If an EWS indicator is affected by the processing lab, then the usefulness of the indicator is rather limited, no?

Answer: The higher-order statistics that are computed to obtain the different EWS indicators can be influenced by the processing of the raw ice core data to derive the final time series. This can lead to biases and, hence, to a masking of an underlying signal of critical slowing down and associated EWS. The degree to which this occurs depends on the exact preprocessing conducted for each core, and therefore we cannot expect to obtain the same EWS for each core. Of course, in an ideal setting, the signal-to-noise ratio would be so high that these effects would not matter. But in the case of data from different ice cores, processed differently in different labs, that cannot be expected. This is, in our opinion, not a matter of the usefulness of the indicator, but rather reflects the impact of the underlying uncertainties.

**Changes:**

• added in I.555: ", because the higher-order statistics that are computed to obtain the different EWS indicators can be influenced by the processing of the raw ice core data to derive the final time series. This can lead to biases and, hence, to a masking of an underlying signal of critical slowing down and associated EWS. The degree to which this occurs depends on the exact preprocessing conducted for each core, and therefore we cannot expect to obtain the same EWS for different ice cores, processed differently in different labs. We further argue that this does not yield a limitation of the usefulness of EWS indicators, but rather reflects the impact of the underlying uncertainties."

Finally, to aid the overall flow and the interpretation of the results in their physical context, I would suggest to switch Section 4.2 and 4.1. That would result in a much stronger ending, than the discussion of possible ice core differences.

Answer: We agree and will switch Sections 4.2 and 4.1 in the revised manuscript.

**Changes:**

• The sections have been switched: Sect. 4.1 Differences between ice core records, starting in I. 496 an Sect. 4.2 Implications of results, starting in I. 564

We highly appreciate the comments above and thank Marlene Klockmann for her suggestions and feedback.

---

## Author Response (AR2)

**Author's response to reports after major revisions of "Inconclusive Early Warning signals for Dansgaard-Oeschger events across Greenland ice cores"**

**Suggestions for revision in Report #1:**

Line 269: "At 95% confidence, one such simultaneous increase is expected (Fig. 3(c) and A3(c))." I find this combination of the 95% confidence threshold and the number of "expected" EWS confusing. My understanding is that, under the null hypothesis, the expected number of (false positive) EWS would be a decimal value slightly greater than zero. This sentence should perhaps instead read "... one such simultaneous increase is statistically significant...".

Answer: We thank the referee for spotting this confusing statement and agree. The sentence has been reformulated as suggested to "At 95% confidence, one such simultaneous increase is statistically significant (Fig. 3(c) and A3(c))."

Line 422: "...whereas GISP2 in 20-year resolution the most EWS...". A word is missing here between "resolution" and "the".

Answer: We corrected this to "...whereas GISP2 in 20-year resolution shows the most EWS...".

Line 438: "There is further no common significant increase in neither the autocorrelation, nor the local Hurst exponent across the different resolutions of the NGRIP record." This sentence is unclear due to the use of a double negative. It would read better if the words "neither" and "nor" were replaced with "either" and "or". Alternatively, this sentence would make better sense if rephrased as follows: "Further, neither the autocorrelation nor the local Hurst exponent show a common significant increase across the different resolutions of the NGRIP record."

Answer: We rephrased this sentence as suggested.

Line 508: "While these discrepancies between the signal do not necessarily influence the EWS considered here, they are remarkable and indicate important regional variations, given their geographical proximity...". This sentence is currently unclear. I would suggest that "signal" be changed to "signals", and that "their geographical proximity" be changed to "the geographical proximity of the ice core sites".

Answer: This sentence has been changed to "While these discrepancies between the signals do not necessarily influence the EWS considered here, they are remarkable and indicate important regional variations, given the geographical proximity of the ice core sites ...".

We thank John Slattery for his comments and the very valuable feedback to this manuscript.

**Suggestions for revision in Report #2:**

I thank the authors for their thorough revisions, the manuscript has greatly improved. I especially appreciate the additional explanation of the method modifications, the comparison to the restoring rate lambda and the new figures 6-8.

Answer: We thank Marlene Klockmann for her comments and the very valuable feedback to this manuscript.

I have no further comments except a very minor one:

I.98-101: it seems that stadials (GS) and interstadials (GI) got confused here? Should it not read "Hence, future changes might be more comparable to past transitions from GI to GS, rather than DO events with changes from GS to GI, during the last glacial period. Past GI-GS transitions, as those shown in Fig. 2, occurred more gradually [...]"? Answer: We agree that GS and GI got confused in this sentence and corrected it accordingly.